# Blocking the Leakage: Manifold-Aware Gradient Projection for Long-Horizon Test-Time Adaptation

**Haoyu Xiong**[1] **Chengchao Wang**[1] **Zhongqiang Wang**[1] **He Huang**[1] **Qiuxia Yang**[2] **Zhengpeng Zhao**[1]
**Yuanyuan Pu**[2,3]

## Abstract

Test-Time Adaptation (TTA) empowers pre-trained models to adapt online to distribution shifts during inference, but such online updates often become unstable in long-horizon deployments. Prevailing approaches attribute this failure to error accumulation from noisy pseudo-labels, relying on heuristics to gate samples used for updates. We argue that this statistical view is insufficient: the problem lies not only in sample quality but also in the directionality of gradients. In this work, we identify a geometric failure mode termed manifold erosion. Through spectral analysis, we find that reliable gradients concentrate in a stable low-rank subspace, while gradients from confident mispredictions are high-rank yet exhibit a persistent directional leakage into this protected subspace. This leakage can accumulate coherently and gradually erode core representations, eventually leading to collapse. To address this, we propose **M**anifold-**A**ware **G**radient **P**rojection (MGP), a geometric intervention that tracks the dominant subspace online and projects gradients onto its orthogonal complement. By blocking the leakage path, MGP decouples stability from plasticity. Extensive experiments on diverse TTA benchmarks demonstrate its long-horizon stability, whereas prior methods often fail.

## 1. Introduction

Deep neural networks achieve excellent performance under the classical i.i.d. assumption that training and test-

[1]School of Information Science and Engineering, Yunnan University, China [2]School of Engineering, Yunnan University, China [3]Key Lab. of IoT Technology and Application in Yunnan Province, China. Correspondence to: Yuanyuan Pu <yuanyuanpu@ynu.edu.cn>.

*Proceedings of the 43rd International Conference on Machine Learning*, Seoul, South Korea. PMLR 306, 2026. Copyright 2026 by the author(s).

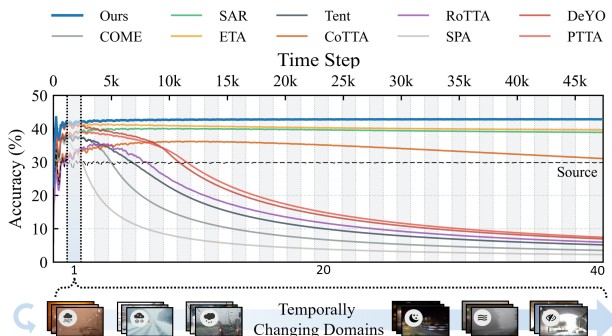

*Figure 1.* **Long-horizon collapse in TTA.** Advanced TTA baselines improve early but tend to collapse in long-horizon test-time adaptation where domains can revisit over time, causing cumulative test-stream accuracy to fall below the source model (dashed line), whereas MGP maintains stable robustness throughout.

ing domains share the same distribution (He et al., 2016; Krizhevsky et al., 2012; Wang et al., 2018). However, this assumption is often violated in real-world deployments: test inputs may drift due to natural corruptions (e.g., noise, blur, weather), sensor shifts, or temporal evolution, and even mild shifts can cause severe performance degradation (Recht et al., 2019). Test-Time Adaptation (TTA) (Wang et al., 2021; Sun et al., 2020) attempts to bridge this gap by updating models online using unlabeled test data. While effective in short episodes, the growing demand for lifelong autonomy raises a critical challenge:

*Can TTA remain stable over long horizons without sacrificing adaptability?*

Current empirical evidence suggests the answer is often negative, particularly under prolonged or more challenging testing scenarios, as shown in Fig. 1 and Table 4. The prevailing view attributes collapse to error accumulation, typically mitigating it by filtering unreliable samples (e.g., confidence/consistency thresholds) (Lee et al., 2024; Niu et al., 2022; Ma et al., 2025) or constraining parameter drift (Niu et al., 2023; Wang et al., 2022). However, we argue that these strategies miss an underlying cause and is insufficient on its own. Filtering restricts the magnitude of updates, but it does not alter the update direction itself. As long as the model makes even a few confidently wrong pre-

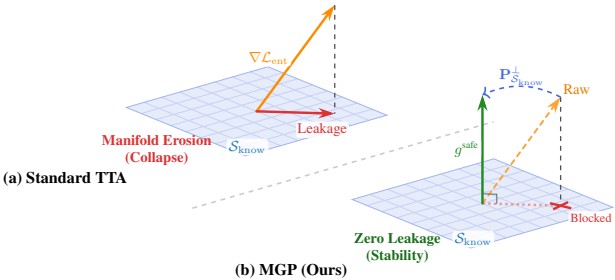

*Figure 2.* **Update geometry comparison**. (a) Standard TTA (Wang et al., 2021) leaks spurious updates into the protected subspace $\mathcal{S}_{\text{know}}$ (solid red). (b) MGP projects onto its orthogonal complement $\mathcal{S}_{\text{know}}^{\perp}$ to block leakage while preserving plasticity.

dictions, the resulting gradients carry a persistent, structural bias. Since filtering cannot correct this geometric misalignment, the bias persists, accumulates, and eventually drives the model to collapse.

In this work, we argue that the cause of collapse is geometric rather than purely statistical. We hypothesize that pre-trained models are spectrally asymmetric: core knowledge concentrates in a low-rank subspace, while adaptation capacity lies in the high-dimensional residual/orthogonal space. Collapse tends to occur when spurious updates, driven by confidently wrong predictions, gradually penetrate this protected subspace. We refer to this mechanism as *manifold erosion*.

To formalize this, we analyze the spectral properties of entropy gradients during adaptation. We decompose the gradient space into a protected (knowledge) subspace $\mathcal{S}_{\text{know}}$ (spanned by dominant directions of reliable gradients) and its orthogonal complement. Through this lens (visualized in Fig. 2), we make the following key observations (see §3):

- **O1:** *Spectral Asymmetry.* Reliable gradients are low-rank and concentrate in a dominant protected subspace $\mathcal{S}_{\text{know}}$, whereas spurious gradients (confident but wrong) exhibit a heavy-tailed, high-rank spectrum.

- **O2:** *Collapse is Directed.* Spurious gradients from confident mispredictions are not isotropic noise. They exhibit a persistent and structured projection onto $\mathcal{S}_{\text{know}}$. This "leakage" accumulates coherently over time, rewriting the model's core features.

- **O3:** *Orthogonal/Residual Preserves Plasticity.* Projecting updates onto the orthogonal complement $\mathcal{S}_{\text{know}}^{\perp}$ effectively blocks the leakage path, preventing collapse while allowing effective adaptation via the high-rank residual space.

Based on these insights, we propose a simple yet effective **M**anifold-**A**ware **G**radient **P**rojection (MGP) scheme for test-time entropy minimization, which focuses on where the optimization moves in parameter space. MGP maintains an online estimate of the protected subspace via robust spectral

tracking and strictly projects gradients onto its orthogonal complement, thereby decoupling stability from plasticity. We support this finding empirically and theoretically:

- **Why? Theoretical Insights (§4.3):** We provide drift bounds showing that stability is determined by the misalignment between the tracked and oracle subspaces. We prove that while unconstrained updates drift linearly with time due to biased leakage ($O(T)$), our projected updates reduce drift to a function of tracking error, which is effectively bounded under spectral separation conditions.

- **When? Long-horizon Robustness (§5):** We subject MGP to rigorous stress tests on diverse synthetic and natural shifts over long horizons (up to 7.4 million samples). MGP maintains stability while matching the plasticity of unconstrained methods. Notably, in a "blind-spot" regime, where the stream consists entirely of samples misclassified by the source model, MGP recovers performance where confidence-based filtering fails completely. This indicates that useful adaptation signal persists in the residual space $\mathcal{S}_{\text{know}}^{\perp}$ even when predictions are wrong.

## 2. Related Work

**Test-Time Adaptation (TTA)** updates a pre-trained model online using unlabeled test data to handle distribution shift, typically via entropy minimization (Wang et al., 2021; Lee et al., 2024). To mitigate instability from online updates, prior work mainly restricts when/what to update: i) confidence/consistency-based sample filtering (Niu et al., 2022; Lee et al., 2024; Niu et al., 2023; Ma et al., 2025; Shamsi et al., 2025; Niu et al., 2025a), ii) binding TTA to the source model via recovery or partial reset mechanisms (Wang et al., 2022; Brahma & Rai, 2023), or iii) teacher/ensemble regularization or augmentation to smooth pseudo-label noise (Yuan et al., 2023; Niu et al., 2025b). Recent studies also explore structural gradient alignment (Shin et al., 2024) or subspace alignment for regression TTA (Adachi et al., 2025). Despite these advances, existing stabilizers mostly act on whether to update (sample gating) or how much to update (regularize/reset), which can be brittle under long-horizon shifts with high-confidence errors. In contrast, MGP stabilizes TTA by controlling where updates occur in parameter space.

**Geometric Adaptation** exploits the geometric structure of representation spaces to bridge distribution gaps. Methods like SSP (Zhu et al., 2024) and SSA (Adachi et al., 2025) construct task-relevant subspaces to align representations or regularize feature distributions. However, these methods address representation geometry rather than directly controlling update directions under online adaptation. In a related vein, ViDA (Liu et al., 2024) separates knowledge into a frozen "homeostatic" branch and a learnable low-rank adapter, using uncertainty to modulate contributions.

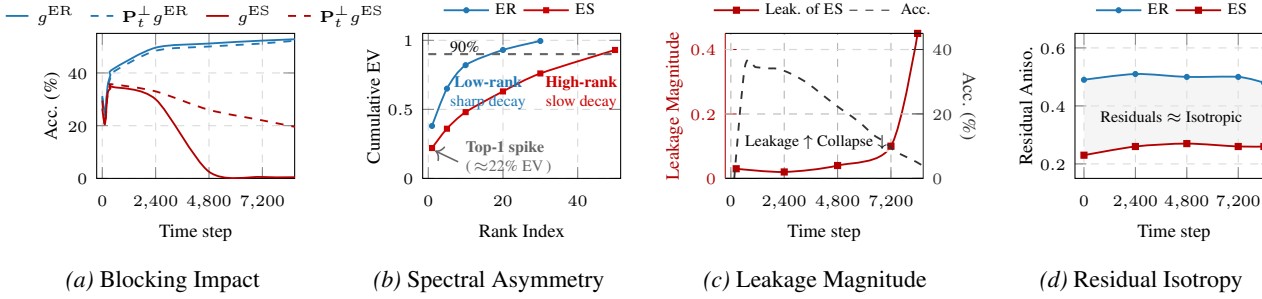

*(a)* Blocking Impact      *(b)* Spectral Asymmetry      *(c)* Leakage Magnitude      *(d)* Residual Isotropy

*Figure 3.* **(a):** Projecting out $\mathcal{S}_t$ prevents ES-induced collapse (dashed red) while preserving adaptation plasticity for ER (dashed blue). **(b):** The cumulative Explained Variance (EV) shows reliable gradients concentrate in a low-rank subspace, whereas spurious gradients have a heavy tail overlapping the protected subspace. **(c):** The projection of ES gradients onto $\mathcal{S}_{\mathrm{know}}$ accumulates coherently over time, correlating with accuracy degradation. **(d):** Gradient residuals in $\mathcal{S}_{\mathrm{know}}^{\perp}$ remain nearly isotropic, serving as a safe buffer for adaptation.

Yet, this modular parameterization does not explicitly regulate optimization geometry, leaving long-horizon adaptation prone to drift and gradual erosion.

## 3. Analysis of TTA Collapse

We develop a mechanistic account of why long-horizon test-time adaptation (TTA) collapses. At time step $t$, given an input $x_t$, standard TTA encourages confident predictions by minimizing the prediction Shannon entropy $\mathcal{L}_{\mathrm{ent}}(x_t; \theta_t) \triangleq -\sum_{c=1}^{C} p_c(x_t; \theta_t) \log p_c(x_t; \theta_t)$, where $p(x_t; \theta_t)$ is the softmax probability over $C$ classes. Thus, the (possibly gated) update direction is

$$g_t^{\mathrm{raw}} \triangleq \nabla_\theta \mathcal{L}_{\mathrm{ent}}(x_t; \theta_t), \qquad g_t \triangleq \mathbb{I}_t \, g_t^{\mathrm{raw}}, \qquad (1)$$

where $\mathbb{I}_t \in \{0, 1\}$ is an update gate used by filtering-based stabilizers (Lee et al., 2024; Niu et al., 2022; 2023). Crucially, the gate $\mathbb{I}_t$ can only scale the update magnitude; it cannot change the update direction. Therefore, long-horizon stability hinges on whether admitted updates carry a biased component along sensitive directions. Entropy minimization indiscriminately increases the leading logit regardless of correctness. Under realistic shifts where confident mispredictions occur, the induced gradients can be systematically destructive, leading to collapse.

Rather than attributing collapse solely to statistical noise accumulation, we suggest the central failure mode is geometric: *collapse is associated with a persistent projection of spurious gradients into a protected, low-rank knowledge subspace, whose coherent accumulation erodes core representations over time.*

To analyze this, we introduce the Prediction Alignment Margin (PAM). For a labeled sample $(x, y)$, define the PAM as $\mathrm{PAM}(x, y; \theta) \triangleq p_y(x; \theta) - \max_{c \neq y} p_c(x; \theta)$.

**Definition 3.1.** Among samples that are actually admitted for updates, i.e., $\mathbb{I}_t = 1$, an admitted sample is Entropy-Reliable (ER) if PAM $> 0$ (admitted and correctly pre-

dicted), and Entropy-Spurious (ES) if PAM $\leq 0$ (admitted but incorrectly predicted).

Based on this separation, we analyze the spectral properties of gradients from ER and ES samples to reveal the geometry of long-horizon adaptation. We detail our key observations **O1–O3** below, supported by evidence in Fig. 3.

### 3.1. Spectral Asymmetry and Leakage

**O1: Reliable gradients are spectrally concentrated.** We hypothesize that model capabilities are spectrally asymmetric, relying on a low-dimensional core subspace. To examine this, we compute the empirical uncentered second-moment matrix over the stream, $\mathbf{C}^{\mathrm{ER}} \triangleq \mathbb{E}[g^{\mathrm{ER}}(g^{\mathrm{ER}})^{\top}]$. As shown in Fig. 3b (blue curve), its spectrum decays sharply, with over 90% of the variance concentrated in a small fraction of the parameter space. This indicates that constructive adaptation lies in a distinct low-rank subspace. We therefore define the *protected subspace* $\mathcal{S}_{\mathrm{know}}$ as the principal subspace spanned by the top-$r$ eigenvectors of $\mathbf{C}^{\mathrm{ER}}$.

**O2: Spurious updates exhibit coherent projection onto** $\mathcal{S}_{\mathrm{know}}$**.** A naive assumption is that gradients from spurious samples ($g^{\mathrm{ES}}$) act as high-rank, isotropic noise that cancels out over time. Our analysis reveals the opposite. While the full spectrum of ES gradients is indeed heavy-tailed (see Fig. 3b, red curve), they exhibit a persistent, nonzero projection onto $\mathcal{S}_{\mathrm{know}}$. We quantify this via the leakage magnitude $\|\mathbf{P}_{\mathcal{S}_{\mathrm{know}}} g_t^{\mathrm{ES}}\|_2$. As visualized in Fig. 3c, this leakage is not random; it accumulates coherently and closely tracks the observed accuracy collapse. This suggests that ES samples are structurally destructive, rewriting core features along the model's most sensitive axes.

**O3: The residual space is sufficient for adaptation.** Does blocking updates to $\mathcal{S}_{\mathrm{know}}$ harm adaptation? We posit that adaptation capacity lies in the orthogonal complement $\mathcal{S}_{\mathrm{know}}^{\perp}$. To validate this, we intervene by projecting updates onto $\mathcal{S}_{\mathrm{know}}^{\perp}$ (Fig. 3a), observing two outcomes:

1. *Stability:* Projecting ES updates (dashed red) significantly mitigates collapse, identifying $\mathcal{S}_{\text{know}}$ as the channel for erosion.

2. *Plasticity:* Projected ER updates (dashed blue) match full-update performance, implying that high-dimensional residual directions are sufficient for adaptation.

Consistently, Fig. 3d shows that gradient residuals in $\mathcal{S}_{\text{know}}^{\perp}$

### 3.2. The Leakage Hypothesis

Combining O1–O3, we formalize the failure mode as *manifold erosion*: for any incoming gradient $g_t$, decompose

$$g_t = \underbrace{\mathbf{P}_{\mathcal{S}_{\text{know}}} g_t}_{\text{leakage into protected subspace}} + \underbrace{\mathbf{P}_{\mathcal{S}_{\text{know}}}^{\perp} g_t}_{\text{residual / plasticity directions}} . \quad (2)$$

Standard TTA collapses because it cannot distinguish these components in the raw stream: even rare but biased leakage into $\mathcal{S}_{\text{know}}$ accumulates continuously over time.

> *Remark* 3.2 (**Why small leakage can dominate over time**). Even if $\|\mathbf{P}_{\mathcal{S}_{\text{know}}} g_t^{\text{ES}}\|_2$ is small per step, what matters is whether it has a nonzero mean (bias) in $\mathcal{S}_{\text{know}}$. High-dimensional components in $\mathbf{P}_{\mathcal{S}_{\text{know}}}^{\perp}$ tend to cancel, while a persistent bias in $\mathcal{S}_{\text{know}}$ accumulates coherently, yielding drift that grows with the horizon.

**Proposition 3.3** (Linear drift under biased leakage). *Fix $\mathcal{S}_{\text{know}}$ and let $\mathbf{P}_{\mathcal{S}_{\text{know}}}$ be its projector. Define the ES-induced protected drift $\Delta_T^{\text{ES}} \triangleq -\eta \sum_{t=0}^{T-1} \mathbb{I}_t \mathbf{P}_{\mathcal{S}_{\text{know}}} g_t^{\text{ES}}$, where $g_t^{\text{ES}}$ denotes the gradient contribution from admitted ES samples. Let $\mu_t^{\text{ES}} \triangleq \mathbb{E}\big[\mathbb{I}_t \mathbf{P}_{\mathcal{S}_{\text{know}}} g_t^{\text{ES}}\big]$. Then $\mathbb{E}[\Delta_T^{\text{ES}}] = -\eta \sum_{t=0}^{T-1} \mu_t^{\text{ES}}$. In particular, if $\mu_t^{\text{ES}} \equiv \mu^{\text{ES}} \neq \mathbf{0}$, then $\mathbb{E}[\Delta_T^{\text{ES}}] = -\eta T \mu^{\text{ES}}$, so the ES-induced protected drift grows linearly with the horizon.*

This highlights why purely filtering whether to update can be insufficient: unless it eliminates the leakage direction, long-horizon erosion remains possible.

### 3.3. From Oracle to Blind Streams: Why $\mathcal{S}_{\text{know}}$ is Trackable without Labels

The analysis above defines $\mathcal{S}_{\text{know}}$ using ER labels, which are unavailable at test time. However, O1 suggests a bridge: the reliable gradient are low-rank and spectrally dominant. Concretely, the raw-stream second moment can be viewed as $\mathbf{C}^{\text{raw}} \approx \alpha \mathbf{C}^{\text{ER}} + (1 - \alpha) \mathbf{C}^{\text{ES}}$, where $\alpha$ is the (unknown) fraction of ER samples. In a standard spiked-covariance view, $\mathbf{C}^{\text{ER}}$ contributes stable low-rank spikes with an eigengap, whereas $\mathbf{C}^{\text{ES}}$ mostly forms a high-dimensional bulk that behaves approximately isotropically in $\mathcal{S}_{\text{know}}^{\perp}$ ( Fig. 3d). Thus, the leading eigenspace of the unlabeled raw stream naturally aligns with $\mathcal{S}_{\text{know}}$, making the protected subspace trackable online without labels.

Therefore, protecting the dominant subspace estimated from raw gradients is empirically sufficient to block the collapse-inducing leakage channel identified above. (We further relate drift inside a reference subspace to functional degradation in Appendix D.9.)

> **Takeaway 1**
>
> TTA collapse is driven by directional leakage of spurious gradients into the low-rank protected subspace $\mathcal{S}_{\text{know}}$. Projecting updates onto the orthogonal complement $\mathcal{S}_{\text{know}}^{\perp}$ blocks this erosion while preserving sufficient plasticity for adaptation.

## 4. Methodology

In §3, we defined a protected subspace $\mathcal{S}_{\text{know}}$. In practice, MGP maintains an online estimate $\mathcal{S}_t$ using recent raw gradients. This yields a two-timescale procedure: (i) a *per-step* projected update that blocks leakage into $\mathcal{S}_t$, and (ii) a *periodic* subspace refresh that tracks slow drift in the dominant gradient directions (Algorithm 1).

### 4.1. Projected Test-Time Entropy Minimization

Formally, let $\theta \in \mathbb{R}^d$ denote model parameters and $g_t^{\text{raw}} = \nabla_\theta \mathcal{L}_{\text{ent}}(x_t; \theta_t) \in \mathbb{R}^d$ the raw entropy gradient. At time step $t$, MGP uses a protected subspace estimate $\mathcal{S}_t \subset \mathbb{R}^d$ spanned by an orthonormal basis $\mathbf{U}_t \in \mathbb{R}^{d \times r_t}$. Define the active orthogonal projector onto $\mathcal{S}_t$ and its residual adaptation projector $\mathbf{P}_t^{\perp}$ (onto the orthogonal complement) as:

$$\mathbf{P}_t \triangleq \mathbf{U}_t \mathbf{U}_t^{\top}, \qquad \mathbf{P}_t^{\perp} \triangleq \mathbf{I} - \mathbf{P}_t. \quad (3)$$

Given the raw gradient $g_t^{\text{raw}} = \nabla_\theta \mathcal{L}_{\text{ent}}(x_t; \theta_t)$, possibly gated by $\mathbb{I}_t$, MGP performs the projected update:

$$\begin{aligned} g_t^{\text{safe}} &:= \mathbf{P}_t^{\perp} g_t^{\text{raw}} = (\mathbf{I} - \mathbf{U}_t \mathbf{U}_t^{\top}) g_t^{\text{raw}}, \\ \theta_{t+1} &= \theta_t - \eta\, \mathbb{I}_t\, g_t^{\text{safe}}. \end{aligned} \quad (4)$$

By construction, this ensures $\mathbf{P}_t(\theta_{t+1} - \theta_t) = \mathbf{0}$. The update strictly preserves the components of $\theta_t$ lying in $\mathcal{S}_t$, forcing all adaptation to occur within the residual space $\mathcal{S}_t^{\perp}$.

### 4.2. Robust Subspace Tracking via Spectral Distillation

Estimating $\mathbf{U}_t$ from noisy, unlabeled test streams is non-trivial. Naive PCA is susceptible to transient spikes. We therefore employ bulk-edge spectral separation: we model the high-dimensional residual component as approximate isotropic noise (the "bulk") and keep only persistent low-rank structure (the "edge").

**Bulk-edge spectral filtering.** We maintain a FIFO buffer $\mathbf{G} \in \mathbb{R}^{n \times d}$ of the recent $n$ raw gradients. To avoid forming the $d \times d$ covariance, we work with the singular values $\{s_i\}_{i=1}^{\min(n,d)}$ of the centered buffer $\mathbf{G}_{\text{cen}}$. Guided

by Marchenko–Pastur (MP) theory (Marchenko & Pastur, 1967) for aspect ratio $\gamma = d/n$, we estimate the noise variance $\hat{\sigma}^2$ via the spectral median and determine the bulk edge $\tau_{\text{edge}}$:

$$\hat{\sigma}^2 = \frac{\text{median}(\{\lambda_i\})}{m_\gamma}, \qquad \tau_{\text{edge}} = \hat{\sigma}^2 \left(1 + \sqrt{\gamma}\right)^2, \quad (5)$$

where $\lambda_i = s_i^2/n$ are covariance eigenvalues and $m_\gamma$ is the theoretical MP median (see Appendix E). We then estimate the spike rank by thresholding above the bulk edge:

$$\hat{r} = \min\left\{r_{\max}, \ \max\left\{1, \ \left|\{i : \lambda_i > \tau_{\text{edge}}\}\right|\right\}\right\}, \quad (6)$$

and take the candidate basis $\tilde{\mathbf{U}}_t \triangleq \mathbf{V}_{:,1:\hat{r}}$. Following the leakage observation (§3.1), we conservatively protect the extracted spike subspace, which blocks both stable "knowledge" directions and consistently aligned contamination.

**Inertial subspace fusion.** Refreshing $\mathbf{U}_t$ by directly setting $\mathbf{U}_t \leftarrow \tilde{\mathbf{U}}_t$ can introduce jitter. We instead update smoothly on the Grassmann manifold by admitting only novel energy outside the current protected subspace. We first residualize the candidate directions:

$$\mathbf{R} = (\mathbf{I} - \mathbf{U}_{t-1}\mathbf{U}_{t-1}^\top)\tilde{\mathbf{U}}_t, \quad (7)$$

and keep columns with sufficiently large residual norm:

$$\mathcal{I}_{\text{novel}} = \left\{j : \ \|\mathbf{R}_{:,j}\|_2 > \tau_{\text{novel}}\|(\tilde{\mathbf{U}}_t)_{:,j}\|_2\right\}. \quad (8)$$

We then augment and re-orthogonalize:

$$\mathbf{U}_t = \text{Truncate}_{r_{\max}}(\text{QR}([\mathbf{U}_{t-1}, \ \mathbf{R}_{:,\mathcal{I}_{\text{novel}}}])), \quad (9)$$

where $\text{Truncate}_{r_{\max}}(\cdot)$ ranks directions by their Rayleigh quotient $u^\top \widehat{\boldsymbol{\Sigma}} u$ under the current buffer covariance $\widehat{\boldsymbol{\Sigma}} = \frac{1}{n}\mathbf{G}_{\text{cen}}^\top \mathbf{G}_{\text{cen}}$ and keeps the top $r_{\max}$. This acts as a geometric low-pass filter, preserving stable directions while reducing sensitivity to transient bulk fluctuations.

### 4.3. Theoretical Guarantees

**What we guarantee.** MGP provides *directional protection*: (i) each update has zero instantaneous drift inside the tracked subspace $\mathcal{S}_t$; (ii) long-horizon drift inside any fixed reference subspace $\mathcal{S}_\star$ is controlled by the misalignment between $\mathcal{S}_t$ and $\mathcal{S}_\star$. All proofs and MP-tracking details are deferred to Appendix D.

**Drift Bounds.** We analyze the update dynamics defined in Eq. (4). For a fixed reference subspace $\mathcal{S}_\star$ with projector $\mathbf{P}_\star$, define the protected drift

$$F_T(\mathcal{S}_\star) \triangleq \|\mathbf{P}_\star(\theta_T - \theta_0)\|_2.$$

---

**Algorithm 1** The proposed MGP algorithm

1: **Input:** pre-trained $\theta_0$, stream $\{x_t\}_{t\geq 0}$
2: **Hyperparams:** lr $\eta$, buffer size $n$, refresh interval $K$, rank bound $r_{\max}$, novelty $\tau_{\text{novel}}$
3: **Init:** $\mathbf{U}_0 \leftarrow \emptyset$, $\mathbf{G} \leftarrow \emptyset$, $\theta_t \leftarrow \theta_0$
4: **for** $t = 0, 1, 2, \ldots$ **do**
5:    $g_t^{\text{raw}} \leftarrow \nabla_\theta \mathcal{L}_{\text{ent}}(x_t; \theta_t)$
6:    $g_t^{\text{safe}} \leftarrow (\mathbf{I} - \mathbf{U}_t\mathbf{U}_t^\top) g_t^{\text{raw}}$
7:    $\theta_{t+1} \leftarrow \theta_t - \eta\, g_t^{\text{safe}}$
8:    $\mathbf{G} \leftarrow \text{FIFOPUSH}(\mathbf{G}, g_t^{\text{raw}})$
9:    **if** $(t + 1) \bmod K = 0$ **then**
10:      $\tau_{\text{edge}} \leftarrow \text{MPBULKEDGE}(\mathbf{G})$       (Eq. (5))
11:      $\tilde{\mathbf{U}}_{t+1} \leftarrow \text{SPIKES}(\mathbf{G}, \tau_{\text{edge}}, r_{\max})$   (Eq. (6))
12:      $\mathbf{U}_{t+1} \leftarrow \text{INERTIALFUSION}(\mathbf{U}_t, \tilde{\mathbf{U}}_{t+1}, \tau_{\text{novel}}, r_{\max})$ (Eq. (7)–Eq. (9))
13:    **else**
14:      $\mathbf{U}_{t+1} \leftarrow \mathbf{U}_t$
15:    **end if**
16: **end for**

---

The key quantity linking tracking quality to stability is the misalignment $\delta_t$ between the projector $\mathbf{P}_t$ (defined in Eq. (3)) and the reference:

$$\delta_t \triangleq \|\mathbf{P}_\star \mathbf{P}_t^\perp\|_2 = \sin\theta_{\max}(\mathcal{S}_\star, \mathcal{S}_t) \in [0, 1],$$

where $\theta_{\max}(\mathcal{S}_\star, \mathcal{S}_t)$ is the largest principal angle of $\mathcal{S}_\star$ relative to $\mathcal{S}_t$ (Appendix D.1, Lemma D.2).

**Lemma 4.1** (Exact tracked-subspace protection). $\mathbf{P}_t(\theta_{t+1} - \theta_t) = \mathbf{0}$ *for all* $t$.

**Theorem 4.2** (Reference-subspace drift bound). *For any fixed* $\mathcal{S}_\star$,

$$F_T(\mathcal{S}_\star) \leq \eta \sum_{t=0}^{T-1} \delta_t \|g_t^{\text{raw}}\|_2.$$

**Beyond worst-case accumulation: sublinear drift under cancellation (and when it fails).** The deterministic bound above can be loose when projected residual updates do not accumulate coherently in protected directions. Let $X_t \triangleq \mathbf{P}_\star \mathbf{P}_t^\perp g_t^{\text{raw}}$ and assume $\|g_t^{\text{raw}}\|_2 \leq G$. Under a martingale-difference condition $\mathbb{E}[X_t \mid \mathcal{F}_{t-1}] = 0$ (Appendix Assumption D.7), Appendix D.5 gives a high-probability sublinear guarantee: for any $\rho \in (0, 1)$,

$$F_T(\mathcal{S}_\star) \leq \eta G \sqrt{2\log(1/\rho)} \left(\sum_{t<T} \delta_t^2\right)^{1/2} = O\left((\sup_{t<T} \delta_t)\sqrt{T}\right)$$

In contrast, any persistent bias in protected directions yields $\Theta(T)$ drift in expectation (Proposition 3.3), explaining why even small leakage can dominate over long horizons.

**Why MP spectral distillation helps: tracking error $\Rightarrow$ small $\delta_t$.** Appendix D.8 shows that under a local spiked-covariance model with eigengap gap separating spikes from the MP bulk edge, MP thresholding + Davis–Kahan controls misalignment (see Theorem D.23):

$$\delta(\mathcal{S}_\star, \widehat{\mathcal{S}}) \leq \frac{2\|\widehat{\boldsymbol{\Sigma}} - \boldsymbol{\Sigma}\|_2}{\text{gap}}.$$

Thus, long-horizon stability is tracking-limited: better co-variance estimation (larger buffer $n$ / larger eigengap) implies smaller $\delta_t$ and hence smaller $F_T(\mathcal{S}_\star)$.

> **Takeaway 2**
>
> MGP blocks tracked leakage directions exactly (Lemma 4.1); collapse-inducing drift inside a fixed "knowledge" subspace can only arise through mis-tracking ($\delta_t$) or coherent bias.

## 5. Experiments

We carry out extensive experiments to answer the following questions: **Q1:** Can the method avoid collapse in long-horizon test-time adaptation? (§5.2); **Q2:** Can it adapt when confidence is systematically misleading (blind-spot regime)? (§5.3); **Q3:** Do dynamics match leakage/projection diagnosis and tracking-limited theory? (§5.4).

### 5.1. Setup

**Benchmarks and Baselines.** We conduct experiments on widely used and representative TTA benchmarks: 1) *Synthetic distribution shifts*, including ImageNet-C (Hendrycks & Dietterich, 2019), ImageNet-3DCC (Kar et al., 2022), and ImageNet-C-Bar (Mintun et al., 2021); and 2) *Natural distribution shifts*, such as ImageNet-R (Hendrycks et al., 2021) and ImageNet-K (Wang et al., 2019). To rigorously stress-test stability, we evaluate under a long-horizon cyclic revisits protocol that extends the continually evolving environment of Wang et al. (2022) indefinitely, setting the horizon to 40 rounds (i.e., 600 dynamic corruptions on ImageNet-C), which is empirically sufficient to expose degradation trends and model collapse. We compare MGP against official implementations of state-of-the-art baselines: TENT (Wang et al., 2021), ETA (Niu et al., 2022), SAR (Niu et al., 2023), CoTTA (Wang et al., 2022), RoTTA (Yuan et al., 2023), TRIBE (Su et al., 2024), DeYO (Lee et al., 2024), SPA (Niu et al., 2025b), ViDA (Liu et al., 2024), PTTA (Ma et al., 2025), and COME (Zhang et al., 2025).[1] PTTA is implemented based on DeYO for comparison.

*Table 1.* GPU runtime (s) on ImageNet-C (1,000 images, ResNet-50).

| Method | Time (s) |
|--------|----------|
| Source | 0.68 |
| TENT | 1.30 |
| ETA | 1.37 |
| CoTTA | 23.21 |
| RoTTA | 31.10 |
| SAR | 2.77 |
| DeYO | 1.95 |
| TRIBE | 15.19 |
| COME | 1.30 |
| PTTA | 3.61 |
| SPA | 8.84 |
| Ours | 4.28 |

**Models and Implementation Details.** We utilize pre-trained ResNet-50 and ViT-B/16 backbones from `torchvision`. We follow official implementations and

their default optimization settings; unless otherwise specified, we use SGD with momentum 0.9 and a batch size of 64. Learning rates are configured as $2.5 \times 10^{-4}$ for ResNet-50 (on synthetic shifts) and $1.0 \times 10^{-3}$ for ViT-B/16 (on natural shifts). Regarding MGP, we set the buffer size $n$ and maximum rank $r_{\max}$ to 32, with an update interval $K$ of 100. Following TENT (Wang et al., 2021), we restrict adaptation to the normalization layers. As shown in Table 1, our projection incurs moderate overhead ($O(d\, r_t)$ per step). All reported results are averaged over 3 random seeds. Further details are provided in Appendix A.

### 5.2. Stability to Long-Horizon Cyclic Revisits

**Intra-Dataset Revisits.** Across per-benchmark revisit streams on both synthetic (Table 2, left) and natural shifts (Table 3), many TTA baselines exhibit early gains followed by late drift. On synthetic corruptions like IN-C, unconstrained entropy updates lead to collapse by Round 40 (e.g., TENT/DeYO/PTTA drop to $< 1\%$). This fragility is exacerbated on natural shifts; on ImageNet-R, DeYO and PTTA suffer severe forgetting ($\Delta = -48.32\%$ and $-46.50\%$). MGP preserves performance throughout the entire stream, and shows superior plasticity on IN-R (+4.50%), consistent with the bounded-forgetting mechanism in Theorem 4.2.

**Inter-Dataset Revisits.** To go beyond per-benchmark cycles and rule out dataset-specific effects, we evaluate a heterogeneous stream cycling across ImageNet* families (IN-C $\rightarrow$ IN-3DCC $\rightarrow$ IN-C-Bar, Table 2, right). This regime requires retaining stable knowledge while accommodating distinct domain variations. Existing stabilization methods succumb to severe degradation, e.g., SAR, $\Delta = -37.75\%$ and PTTA, $\Delta = -42.57\%$. MGP demonstrates superior robustness with $\Delta = +1.45\%$, supporting the subspace concentration hypothesis: by restricting updates to $\mathcal{S}_t^\perp$, MGP decouples knowledge preservation from adaptation.

### 5.3. Resilience to Blind-Spot Streams

To probe failure modes where confidence is systematically misleading, we construct a blind-spot regime where the stream consists exclusively of samples initially misclassified by the source model. In this regime, entropy is a spurious proxy for correctness.

Results in Table 4 reveal a clear contrast: methods relying on entropy or confidence heuristics struggle severely. On IN-K and IN-R, most baselines fail completely (e.g., Acc. $\approx 0\%$). Similarly, on IN-C, 6 out of 10 methods suffer a degradation of over 30 percentage points in accuracy (e.g., PTTA collapses to $0.48\%$ and DeYO to $0.15\%$). In contrast, MGP effectively exploits residual adaptation signal. On IN-C, it achieves a final accuracy of 39.69%. On natural shifts where baselines fail, MGP achieves notable recovery: 27.01% on IN-K and 32.85% on IN-R. This supports the

---

[1]RoTTA and TRIBE are limited to BatchNorm-based models, whereas ViDA is limited to LayerNorm-based models.

*Table 2.* Comparisons with SOTAs on ImageNet (IN) -C/-3DCC/-C-Bar under long-horizon cyclic revisits w.r.t. average accuracy (%) at **R**ound 1/20/40, and $\Delta = $ R40 $-$ R1. **Left:** Intra-dataset revisits (cycles within the same dataset). **Right:** Inter-dataset revisits (cycles across different datasets). Best results are **bolded**. $\Delta$ values are color-coded: gains in blue, losses in red.

| Method | IN-C | | | | IN-3DCC | | | | IN-C-Bar | | | | IN-C $\to$ 3DCC $\to$ C-Bar | | | |
|---|---|---|---|---|---|---|---|---|---|---|---|---|---|---|---|---|
| | Revisit $\longrightarrow$ | | | | Revisit $\longrightarrow$ | | | | Revisit $\longrightarrow$ | | | | Revisit $\longrightarrow$ | | | |
| | **R1** | **R20** | **R40** | $\Delta$ | **R1** | **R20** | **R40** | $\Delta$ | **R1** | **R20** | **R40** | $\Delta$ | **R1** | **R20** | **R40** | $\Delta$ |
| Source | 7.19 | 7.19 | 7.19 | 0.0 | 14.35 | 14.35 | 14.35 | 0.0 | 10.75 | 10.75 | 10.75 | 0.0 | 10.76 | 10.76 | 10.76 | 0.0 |
| TENT ICLR 21 | 37.51 | 0.47 | 0.46 | -37.05 | 39.41 | 1.55 | 0.56 | -38.85 | 43.03 | 19.36 | 16.17 | -26.86 | 37.49 | 0.49 | 0.49 | -37.00 |
| ETA ICML 22 | 41.07 | 39.31 | 38.72 | -2.35 | 41.49 | 39.77 | 38.87 | -2.62 | 44.78 | 44.17 | 43.56 | -1.22 | 40.80 | 36.59 | 36.07 | -4.73 |
| CoTTA CVPR 22 | 31.80 | 31.37 | 22.58 | -9.22 | 34.60 | 34.67 | 34.67 | +0.07 | 39.01 | 39.01 | 39.03 | +0.02 | 39.74 | 17.37 | 19.68 | -20.06 |
| RoTTA CVPR 23 | 31.28 | 0.65 | 0.19 | -31.09 | 36.23 | 1.83 | 0.53 | -35.70 | 37.68 | 0.61 | 0.71 | -36.97 | 40.29 | 0.12 | 0.12 | -40.17 |
| SAR ICLR 23 | 38.31 | 38.35 | 37.47 | -0.84 | 40.90 | 40.76 | 40.74 | -0.16 | 42.94 | 45.07 | 44.84 | +1.90 | 38.12 | 0.52 | 0.37 | -37.75 |
| DeYO ICLR 24 | 40.94 | 0.14 | 0.13 | -40.81 | 41.33 | 1.37 | 0.27 | -41.06 | 44.00 | 39.77 | 38.65 | -5.35 | 36.78 | 17.19 | 5.99 | -30.79 |
| TRIBE AAAI 24 | 32.41 | 24.85 | 24.80 | -7.61 | 39.95 | 27.45 | 27.43 | -12.52 | 38.76 | 30.69 | 30.01 | -8.75 | 36.23 | 1.56 | 0.45 | -35.78 |
| COME ICLR 25 | 36.95 | 0.57 | 0.42 | -36.53 | 39.19 | 0.79 | 0.69 | -38.50 | 42.82 | 1.29 | 0.98 | -41.84 | 32.41 | 25.07 | 25.19 | -7.22 |
| PTTA ICML 25 | **42.50** | 35.95 | 0.42 | -42.08 | 41.82 | 39.04 | 34.26 | -7.56 | **45.79** | 44.31 | 43.23 | -2.56 | **42.66** | 0.16 | 0.09 | -42.57 |
| SPA ICML 25 | 36.29 | 0.84 | 0.61 | -35.68 | 37.51 | 0.10 | 0.10 | -37.41 | 40.79 | 0.17 | 0.12 | -40.67 | 23.27 | 0.16 | 0.16 | -23.11 |
| **Ours** | 41.90 | **42.88** | **42.96** | +1.06 | 41.88 | **42.44** | **42.43** | +0.55 | 45.27 | **46.78** | **46.72** | +1.45 | 41.63 | **43.08** | **43.08** | +1.45 |

*Table 3.* Comparison with SOTAs on natural shifts (ImageNet (IN)-K/-R) under long-horizon cyclic revisits w.r.t. average accuracy (%) at **R**ound 1/20/40 and $\Delta = $ R40 $-$ R1.

| Method | IN-K | | | | IN-R | | | |
|---|---|---|---|---|---|---|---|---|
| | Revisit $\longrightarrow$ | | | | Revisit $\longrightarrow$ | | | |
| | **R1** | **R20** | **R40** | $\Delta$ | **R1** | **R20** | **R40** | $\Delta$ |
| Source | 29.95 | 29.95 | 29.95 | 0.00 | 44.44 | 44.44 | 44.44 | 0.00 |
| TENT ICLR 21 | 29.98 | 0.17 | 0.10 | -29.88 | 22.26 | 0.49 | 0.47 | -21.79 |
| ETA ICML 22 | 41.42 | 0.25 | 0.21 | -41.21 | 55.46 | 48.28 | 35.26 | -20.20 |
| CoTTA CVPR 22 | 29.94 | 30.06 | 30.02 | +0.08 | 44.44 | 44.48 | 44.42 | -0.02 |
| SAR ICLR 23 | 33.41 | 31.10 | 27.65 | -5.76 | 46.90 | 44.82 | 40.41 | -6.49 |
| DeYO ICLR 24 | 32.65 | 0.92 | 0.92 | -31.73 | 48.81 | 0.60 | 0.49 | -48.32 |
| ViDA ICLR 24 | 15.20 | 0.10 | 0.10 | -15.10 | 34.57 | 0.46 | 0.46 | -34.11 |
| COME ICLR 25 | 20.65 | 0.10 | 0.10 | -20.55 | 42.40 | 8.48 | 1.67 | -40.73 |
| PTTA ICML 25 | 28.17 | 1.04 | 0.69 | -27.48 | 47.68 | 14.88 | 1.18 | -46.50 |
| SPA ICML 25 | 39.12 | 0.14 | 0.13 | -38.99 | 43.43 | 0.65 | 0.64 | -42.79 |
| **Ours** | **43.41** | **45.96** | **45.94** | +2.53 | **57.01** | **61.46** | **61.51** | +4.50 |

*Table 4.* Comparison with SOTAs on synthetic (IN-C) and natural shifts (IN-K, IN-R) under blind-spot long-horizon cyclic revisits w.r.t. average accuracy (%) at **R**ound 1/40 and $\Delta = $ R40 $-$ R1.

| Method | IN-C | | | IN-K | | | IN-R | | |
|---|---|---|---|---|---|---|---|---|---|
| | Revisit $\longrightarrow$ | | | Revisit $\longrightarrow$ | | | Revisit $\longrightarrow$ | | |
| | **R1** | **R40** | $\Delta$ | **R1** | **R40** | $\Delta$ | **R1** | **R40** | $\Delta$ |
| Source | 0.00 | 0.00 | 0.00 | 0.00 | 0.00 | 0.00 | 0.00 | 0.00 | 0.00 |
| TENT ICLR 21 | 34.78 | 0.59 | -34.19 | 0.60 | 0.03 | -0.57 | 1.06 | 0.34 | -0.72 |
| ETA ICML 22 | 38.07 | 35.95 | -2.12 | 20.60 | 0.12 | -20.48 | 19.78 | 0.79 | -18.99 |
| CoTTA CVPR 22 | 28.76 | 20.27 | -8.49 | 0.02 | 0.03 | 0.01 | 0.04 | 0.04 | 0.00 |
| RoTTA CVPR 23 | 35.76 | 0.27 | -35.49 | – | – | – | – | – | – |
| SAR ICLR 23 | 35.23 | 29.63 | -5.60 | 2.08 | 1.64 | -0.44 | 2.21 | 1.55 | -0.66 |
| DeYO ICLR 24 | 37.90 | 0.15 | -37.75 | 2.98 | 0.24 | -2.74 | 0.70 | 0.34 | -0.36 |
| TRIBE ICLR 24 | 33.36 | 13.34 | -20.02 | – | – | – | – | – | – |
| ViDA ICLR 24 | – | – | – | 0.70 | 0.03 | -0.67 | 1.24 | 0.34 | -0.90 |
| COME ICLR 25 | 37.91 | 0.61 | -37.30 | 0.78 | 0.03 | -0.75 | 1.47 | 1.67 | +0.20 |
| PTTA ICML 25 | 35.53 | 0.48 | -35.05 | 0.15 | 0.03 | -0.12 | 0.74 | 0.34 | -0.40 |
| SPA ICML 25 | 33.29 | 0.23 | -33.06 | 1.63 | 0.36 | -1.27 | 1.00 | 0.35 | -0.65 |
| **Ours** | **38.87** | **39.69** | +0.82 | **22.97** | **27.01** | +4.04 | **25.29** | **32.85** | +7.56 |

claim that even when predictions are wrong, gradients contain useful residuals in $\mathcal{S}_t^\perp$. Projecting out the leakage in $\mathcal{S}_t$ allows MGP to safely learn from hard samples.

### 5.4. Additional Discussions

We now connect the leakage mechanism in Sec. 3 to the tracking-limited stability predicted by Theorem 4.2. Fig. 4 summarizes: (a) structured leakage alignment, (b) tracker–oracle fidelity, (c) residual anisotropy in the projected space, and (d) reference-subspace drift together with accuracy.

**Leakage is Structured, not Random (Fig. 4a).** We fix a reference subspace $\mathcal{S}_0$ distilled at initialization with orthonormal basis $\mathbf{B}$ and projector $\mathbf{P}_{\mathcal{S}_0}$. For any gradient $g$, we measure its alignment to $\mathcal{S}_0$ as $a(g; \mathcal{S}_0) \triangleq \frac{\|\mathbf{B}^\top g\|_2}{\|g\|_2}$. We report $a^{\mathrm{ER}} = a(g^{\mathrm{ER}}; \mathcal{S}_0)$, $a^{\mathrm{ES}} = a(g^{\mathrm{ES}}; \mathcal{S}_0)$, and a random-direction baseline $a^{\mathrm{rand}}$. We observe that spurious gradients exhibit a nontrivial projection onto $\mathcal{S}_0$ (well above

$a^{\mathrm{rand}}$), indicating that collapse is driven by directional components that can accumulate coherently over time, rather than by isotropic noise that averages out.

**Raw Gradients Suffice to Track the Oracle Structure (Fig. 4b).** A core premise of MGP is that the unlabeled raw stream reveals the protected structure. We validate this by comparing the misalignment between the tracked subspace $\mathcal{S}_t$ (from raw gradients) and the oracle protected subspace $\mathcal{S}_{\mathrm{know}}$ (from labeled ER): $\delta_t^{\mathrm{know}} \triangleq \sin\theta_{\max}(\mathcal{S}_{\mathrm{know}}, \mathcal{S}_t)$. MGP maintains $\delta_t^{\mathrm{know}}$ low throughout adaptation, supporting the mixture view in § 3.3 that the leading eigenspace of raw-gradient covariance aligns with $\mathcal{S}_{\mathrm{know}}$ (More results across architectures and datasets in Appendix C.3).

**Residual Updates Remain Incoherent (Fig. 4c).** MGP restricts adaptation to $\mathcal{S}_t^\perp$. We quantify how structured these residual updates are by the residual anisotropy $\nu_t$ (std/mean of residual eigenvalues, see Appendix A.4). The observed

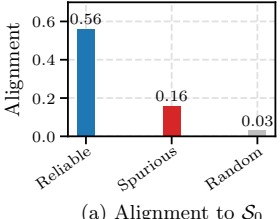 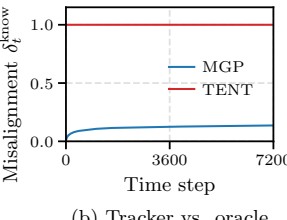 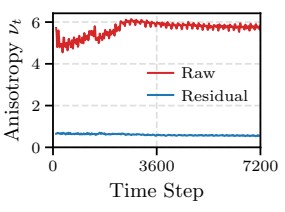 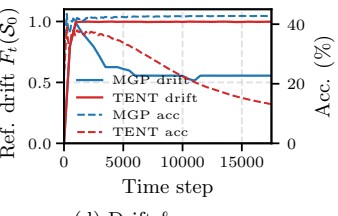

(a) Alignment to $\mathcal{S}_0$    (b) Tracker vs. oracle    (c) Residual anisotropy    (d) Drift & accuracy

*Figure 4.* **Spectral diagnostics on ImageNet-C. (a)** Spurious gradients (ES) show nontrivial alignment with the reference protected subspace $\mathcal{S}_0$ compared to random vectors. **(b)** The unsupervised subspace $\mathcal{S}_t$ closely tracks the oracle $\mathcal{S}_{\text{know}}$ (low misalignment $\delta_t^{\text{know}}$). **(c)** Projected updates in $\mathcal{S}_t^{\perp}$ maintain low anisotropy, avoiding coherent drift. **(d)** MGP bounds the parameter drift within $\mathcal{S}_0$ (solid line), preventing the accuracy collapse (dashed) seen in TENT.

low anisotropy (typically $\nu_t < 0.6$) suggests that $\mathcal{S}_t^{\perp}$ acts as a benign high-dimensional buffer in which updates are less prone to coherent accumulation.

**Drift inside Reference Subspace Predicts Collapse (Fig. 4d).** To measure erosion of protected directions, we track drift within $\mathcal{S}_0$: $F_t(\mathcal{S}_0) \triangleq \|\mathbf{P}_{\mathcal{S}_0}(\theta_t - \theta_0)\|_2$. For unconstrained entropy minimization (Wang et al., 2021), $F_t(\mathcal{S}_0)$ grows and coincides with accuracy collapse. In contrast, MGP keeps $F_t(\mathcal{S}_0)$ bounded and avoids collapse, aligning with Theorem 4.2: long-horizon stability is achieved by suppressing shifts in the protected subspace.

**Stress Tests under Controlled Assumption Violations.** We evaluate robustness when two modeling conditions are intentionally weakened: (i) *structured residuals* via rank-1 anisotropy injected into the update residuals (larger $\nu$), and (ii) *tracking lag* via a larger refresh interval $K$ (more misalignment). Table 5 shows that these violations degrade performance gracefully (moderate accuracy drops without catastrophic collapse), consistent with a tracking-limited failure mode where degradation appears mainly as increased tracking error and drift, rather than unbounded erosion.

*Table 5.* **Stress tests under controlled violations.** We vary (i) residual anisotropy via rank-1 injection strength $\kappa$ and (ii) tracking quality via refresh interval $K$. $\Delta$ Acc. denotes final accuracy change vs. default in Table 2. $\nu$ and $\delta$ are the measured anisotropy and misalignment metrics (Details in Appendix A.4).

| Factor | Condition | Observed | $\Delta$ Acc. |
|---|---|---|---|
| Residual structure | Low anisotropy ($\kappa = 0.5$) | $\nu \approx 0.72$ | -0.14% |
| | High anisotropy ($\kappa = 100$) | $\nu \approx 8.26$ | -1.80% |
| Tracking frequency | Frequent refresh ($K = 10$) | $\delta \approx 0.13$ | -0.46% |
| | Sparse refresh ($K = 1000$) | $\delta \approx 0.28$ | -1.89% |

**Hyperparameter Sensitivity.** As shown in Fig. 5, stability persists across a wide range of configurations, provided the protected rank avoids under-protection (too small $r_{\text{max}}$). Moreover, fixed-rank baselines are sensitive to rank (too small misses leakage; too large suppresses residual plasticity), whereas MGP automatically selects the effective rank via MP bulk-edge separation, achieving near-optimal performance without tuning. With a sufficient buffer (e.g., $n = 32$), performance peaks at moderate refresh intervals

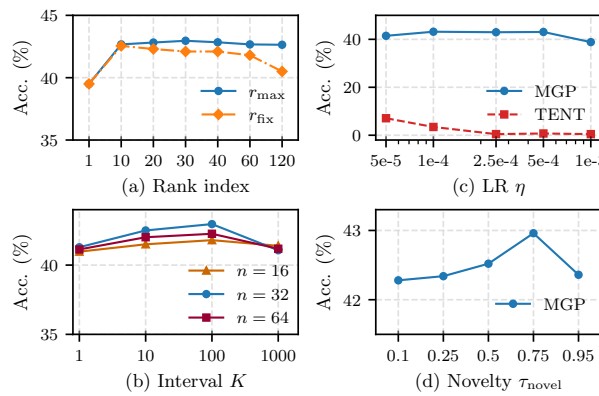

(a) Rank index    (c) LR $\eta$

(b) Interval $K$    (d) Novelty $\tau_{\text{novel}}$

*Figure 5.* Effect of hyperparameters in MGP. We vary (a) maximum rank $r_{\text{max}}$ vs. fixed rank $r_{\text{fix}}$, (b) update interval $K$ (buffer size $n$), (c) learning rate $\eta$, and (d) novelty threshold $\tau_{\text{novel}}$ under long-horizon ImageNet-C revisits (final accuracy, %).

(e.g., $K = 100$) and a novelty threshold $\tau_{\text{novel}} = 0.75$, supporting the assumption that the dominant subspace evolves slowly and needs balanced admission of new directions.

# 6. Conclusion

This work introduces a geometric framework, supported by spectral analysis, to ensure the long-horizon stability of TTA, motivated by the hypothesis that pre-trained models are susceptible to manifold erosion. We argue that model collapse is not only a statistical artifact of noisy pseudo-labels, but also a geometric failure in which spurious gradients penetrate the low-rank subspace hosting core knowledge. We formalize this intuition via the leakage hypothesis and show that MGP decouples stability from plasticity by projecting updates onto the orthogonal residual space (Sec. 4). Spectral diagnostics indicate that reliable knowledge concentrates in a protected subspace and that MGP enforces a geometric barrier against coherent drift (Sec. 3). Extensive experiments across representative TTA benchmarks verify that MGP can effectively prevent collapse while preserving adaptation plasticity, offering a geometric solution to the instability of online entropy minimization.

## Acknowledgements

This work was supported by the National Natural Science Foundation of China (61271361, 61761046, 62162068, 52102382, and 62362070); the Applied Basic Research Program of Yunnan Province (202001BB050043, 202401AS070149); the Major Science and Technology Special Project of Yunnan Province (202302AF080006); the Yunnan University School of ISE Graduate Student Scientific Research Project (ISEGS2026A03, ISEGS2026B03); the Yunnan Provincial Department of Education Science Research Fund (2026Y0180); the Yunnan Key Laboratory of Low-light Night Vision Technology and Intelligent Visual Navigation (202449CE340004); and the Open Research Project of the Yunnan University Resilience and Excellence Children's Character Development Platform (K207003240021).

## Impact Statement

This paper presents work whose goal is to advance the field of Machine Learning. There are many potential societal consequences of our work, none of which we feel must be specifically highlighted here.

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

# APPENDIX

**Contents**

# A. More Implementation Details

## A.1. More Details on Datasets

**Synthetic corruption benchmarks.** We evaluate on three complementary corruption benchmarks derived from ImageNet-1K. ImageNet-C[2] (Hendrycks & Dietterich, 2019) applies 15 common corruptions (grouped into noise, blur, weather, and digital categories) at 5 severity levels to the 50K validation images; following prior continual TTA work, we use severity 5 and treat each corruption as one domain, so one round visits 15 domains and 40 rounds yield 600 domain visits. ImageNet-3DCC[3] (Kar et al., 2022) introduces 21 geometry-aware corruptions across 8 families (depth-of-field, camera motion, lighting, video, weather, view changes, semantics, and noise) that depend on depth and 3D cues, better reflecting real-world distribution shifts. ImageNet-C-Bar[4] (Mintun et al., 2021) provides 10 algorithmically selected corruptions that are perceptually dissimilar from ImageNet-C yet remain natural and human-interpretable, serving as an out-of-distribution stress test for methods tuned on standard corruptions.

**Corruption order used in the main experiments.** For reproducibility, we list the exact corruption order used to construct each round of the main synthetic streams. For the inter-dataset protocol in Table 2, we cycle ImageNet-C → ImageNet-3DCC → ImageNet-C-Bar, and within each benchmark we preserve the following fixed order:

- **ImageNet-C:** motion blur, snow, fog, shot noise, defocus blur, contrast, zoom blur, brightness, frost, elastic transform, glass blur, gaussian noise, pixelate, jpeg compression, impulse noise.

- **ImageNet-3DCC:** flash, h265 abr, far focus, bit error, low light, z motion blur, fog 3d, near focus, iso noise, h265 crf, xy motion blur, color quant.

- **ImageNet-C-Bar:** blue noise, brownish noise, caustic refraction, checkerboard cutout, cocentric sine waves, inverse sparkles, perlin noise, plasma noise, single frequency greyscale, sparkles.

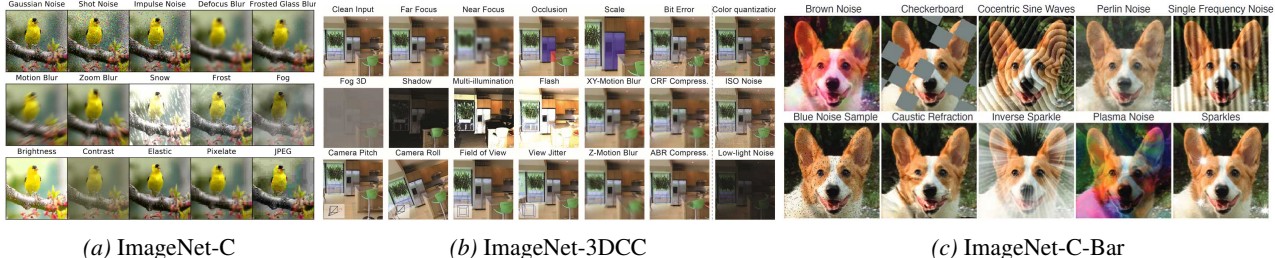

*(a)* ImageNet-C  *(b)* ImageNet-3DCC  *(c)* ImageNet-C-Bar

*Figure 6.* Examples from corruption benchmarks

**Natural distribution shift benchmarks.** We additionally evaluate on natural shifts that do not correspond to synthetic pixel corruptions. ImageNet-R[5] (Hendrycks et al., 2021) contains artistic renditions of ImageNet classes; and ImageNet-K (ImageNet-Sketch)[6] (Wang et al., 2019) is a sketch/abstraction-style shift. These datasets stress semantic/style shifts where confidence can be misleading and collapse is more likely under long-horizon continual adaptation.

## A.2. Details on Evaluation Protocols

We formally define the data streams and evaluation protocols used in our experiments. Let $f_{\theta_t}$ denote the model at time step $t$ with parameters $\theta_t$, adapting to a stream of unlabeled samples $x_t$.

---

[2]https://github.com/hendrycks/robustness
[3]https://github.com/EPFL-VILAB/3DCommonCorruptions
[4]https://github.com/facebookresearch/augmentation-corruption
[5]https://github.com/hendrycks/imagenet-r
[6]https://github.com/HaohanWang/ImageNet-Sketch

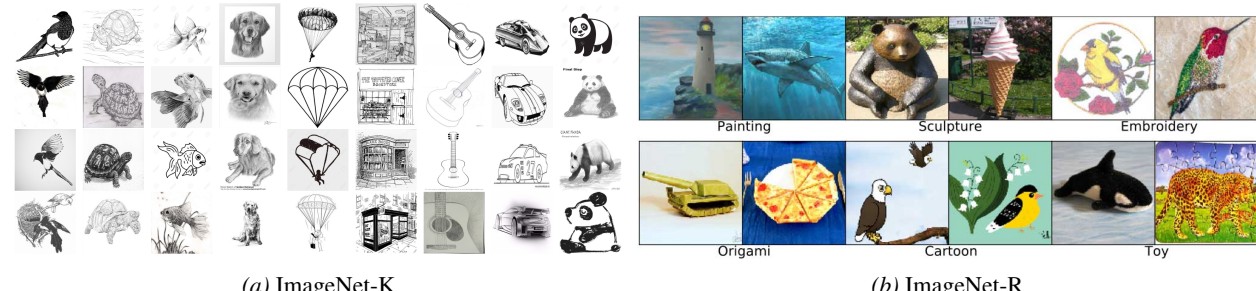

| *(a)* ImageNet-K | *(b)* ImageNet-R |

*Figure 7.* Examples from natural distribution shift benchmarks

**Intra-dataset revisits.** We define a base sequence of distribution shifts $\mathcal{Q} = (\mathcal{D}_1, \mathcal{D}_2, \ldots, \mathcal{D}_M)$, where each $\mathcal{D}_k$ represents a distinct corruption type or domain (e.g., for ImageNet-C, 15 corruption types at Severity 5). A single round $r$ corresponds to the sequential processing of $\mathcal{Q}$. To simulate lifelong non-stationary adaptation, the full evaluation stream $\mathcal{S}_{\text{intra}}$ is constructed by cyclically repeating $\mathcal{Q}$ for a total of $R$ rounds:

$$\mathcal{S}_{\text{intra}} = \underbrace{[\mathcal{D}_1, \ldots, \mathcal{D}_M]}_{\text{Round 1}} \oplus \underbrace{[\mathcal{D}_1, \ldots, \mathcal{D}_M]}_{\text{Round 2}} \oplus \cdots \oplus \underbrace{[\mathcal{D}_1, \ldots, \mathcal{D}_M]}_{\text{Round } R} \tag{10}$$

where $\oplus$ denotes stream concatenation. We set $R = 40$ for all experiments (e.g., total $15 \times 40 = 600$ sub-tasks for ImageNet-C). We denote the model accuracy at the end of round $r$ as $\text{Acc}_r$. The stability metric is defined as the performance delta over the adaptation horizon: $\Delta = \text{Acc}_R - \text{Acc}_1$.

**Inter-dataset revisits.** To assess robustness against heterogeneous shifts, we construct a compound stream from a set of diverse benchmark datasets $\mathbb{B} = \{\mathcal{B}_C, \mathcal{B}_{3D}, \mathcal{B}_{\text{Bar}}\}$, corresponding to ImageNet-C, ImageNet-3DCC, and ImageNet-C-Bar, respectively. Unlike intra-dataset revisits where distributions $P(\mathcal{D}_k)$ share the same generation logic (e.g., 2D noise), here the transition $P(\mathcal{B}_i) \to P(\mathcal{B}_j)$ implies a fundamental change in the shift mechanism. The evaluation stream $\mathcal{S}_{\text{inter}}$ is defined as the cyclic repetition of these benchmarks:

$$\mathcal{S}_{\text{inter}} = \bigotimes_{r=1}^{R} (\mathcal{B}_C \oplus \mathcal{B}_{3D} \oplus \mathcal{B}_{\text{Bar}}) \tag{11}$$

where $\bigotimes_{r=1}^{R}$ denotes the repetition of the sequence $R = 40$ times. This protocol tests whether the projected updates in $\mathcal{S}_t^{\perp}$ can accommodate new domain structures without catastrophically overwriting knowledge required for previous domains.

**Blind-spot streams.** This protocol evaluates the method in the "zero-reliability" regime. Let $f_{\theta_0}$ be the source pre-trained model and $\mathcal{D}_{\text{test}}$ be the target dataset (e.g., ImageNet-C). We construct the blind-spot stream $\mathcal{S}_{\text{blind}}$ by filtering $\mathcal{D}_{\text{test}}$ to strictly exclude samples correctly classified by the source model. Formally, let $\hat{y}_0(x) = \arg\max f_{\theta_0}(x)$ be the source prediction and $y$ be the ground truth. The stream is defined as:

$$\mathcal{S}_{\text{blind}} = \{(x, y) \in \mathcal{D}_{\text{test}} \mid \mathbb{1}(\hat{y}_0(x) \neq y) = 1\} \tag{12}$$

Consequently, by construction, the source model has zero accuracy on the adaptation stream itself, i.e.,

$$\text{Acc}(\mathcal{S}_{\text{blind}}; \theta_0) = 0. \tag{13}$$

Since all samples are initially misclassified, confidence-based reliability metrics become uninformative. This setting serves as a stress test for leakage accumulation, as $\mathbb{E}[g^{\text{raw}}]$ is dominated by spurious directions.

### A.3. More Details on Baseline Implementations

**TENT**[7] (Wang et al., 2021) is the foundational TTA method that optimizes the model by minimizing the Shannon entropy of predictions on the test stream. It updates only the affine parameters of the normalization layers while freezing the rest of the network. For optimization, we use SGD for both backbones: for ResNet-50[8] we set the learning rate to $2.5 \times 10^{-4}$, and

---

[7]https://github.com/DequanWang/tent
[8]https://download.pytorch.org/models/resnet50-19c8e357.pth

for ViT-B/16[9] we set the learning rate to $1.0 \times 10^{-3}$. We follow the original TENT procedure and hyperparameters from the official protocol.

**ETA**[10] (**Niu et al., 2022**) improves upon TENT by introducing (i) reliability filtering that discards high-entropy samples using an entropy threshold set to $0.4$ times the natural logarithm of the number of classes, and (ii) redundancy filtering that suppresses samples with similar gradients (similarity threshold $0.05$ on ImageNet-scale settings). For optimization, we use SGD for both backbones: for ResNet-50 we set the learning rate to $2.5 \times 10^{-4}$, and for ViT-B/16 we set the learning rate to $1.0 \times 10^{-3}$. We follow the original ETA procedure and hyperparameters from the official protocol.

**SAR**[11] (**Niu et al., 2023**) stabilizes entropy minimization by combining reliability filtering with Sharpness-Aware Minimization (SAM). It filters out unreliable high-entropy samples and applies SAM on the remaining samples, commonly using perturbation radius $0.05$. SAR further includes a model recovery mechanism that monitors a moving statistic of the entropy objective and resets parameters to the pre-trained state when collapse is detected (triggered when this statistic falls below a small preset threshold). For optimization, we use SGD for both backbones: for ResNet-50 we set the learning rate to $2.5 \times 10^{-4}$, and for ViT-B/16 we set the learning rate to $1.0 \times 10^{-3}$. We follow the default SAR setting with perturbation radius $0.05$ and the authors' recovery rule.

**CoTTA**[12] (**Wang et al., 2022**) stabilizes long-horizon adaptation via a mean-teacher framework and consistency regularization. It maintains a teacher model (an EMA of the student) to generate more stable pseudo-labels (often averaged across augmentations), and trains the student to match the teacher's predictions under test-time augmentations. To mitigate long-term forgetting, CoTTA further uses stochastic restoration, periodically restoring a small fraction of parameters toward the source model. For optimization, we use SGD for both backbones: for ResNet-50 we set the learning rate to $2.5 \times 10^{-4}$, and for ViT-B/16 we set the learning rate to $1.0 \times 10^{-3}$. We follow the original CoTTA procedure and hyperparameters from the official protocol.

**RoTTA**[13] (**Yuan et al., 2023**) targets non-stationary and class-imbalanced streams. It maintains a Category-Balanced Memory Bank (CBM) with a fixed capacity per class (default $64$) and updates with robust pseudo-labeling and prototype refinement to counteract imbalance. For optimization, we use SGD for both backbones: for ResNet-50 we set the learning rate to $2.5 \times 10^{-4}$, and for ViT-B/16 we set the learning rate to $1.0 \times 10^{-3}$. We use the default CBM capacity and update frequency, following the original RoTTA procedure and hyperparameters from the official protocol.

**DeYO**[14] (**Lee et al., 2024**) analyzes entropy minimization failures from a disentangled-factor perspective. It introduces a Pseudo-Label Probability Difference (PLPD) score to separate samples into reliable and unreliable groups, performs entropy minimization mainly on high-PLPD samples, and regularizes updates on uncertain cases to avoid overconfident drift. For optimization, we use SGD for both backbones: for ResNet-50 we set the learning rate to $2.5 \times 10^{-4}$, and for ViT-B/16 we set the learning rate to $1.0 \times 10^{-3}$. We follow the authors' default PLPD threshold (commonly $0.2$) and keep the remaining settings consistent with the official protocol.

**TRIBE**[15] (**Su et al., 2024**) proposes Tri-net self-training with Balanced Normalization for real-world continual TTA. It maintains three roles: an anchor (source model) to preserve source knowledge, a teacher (EMA-smoothed) to provide stable pseudo-labels under augmentation, and a student to be adapted online. TRIBE performs self-training using pseudo-labels selected by confidence/entropy criteria, and introduces Balanced Normalization to counteract class-imbalance and confirmation bias during online updates by rebalancing normalization statistics/features according to predicted class composition. For optimization, we use SGD for both backbones: for ResNet-50 we set the learning rate to $2.5 \times 10^{-4}$, and for ViT-B/16 we set the learning rate to $1.0 \times 10^{-3}$. We follow the official ImageNet hyperparameters, using an entropy threshold of $0.4$ for sample selection and the default balancing/mixing coefficients (both set to $0.5$) with anchor regularization weight $0.1$. We follow the original TRIBE procedure and hyperparameters from the official protocol.

**COME**[16] (**Zhang et al., 2025**) performs test-time adaptation by conservatively minimizing entropy to avoid collapse from

---

[9] https://download.pytorch.org/models/vit_b_16-c867db91.pth
[10] https://github.com/mr-eggplant/EATA
[11] https://github.com/mr-eggplant/SAR
[12] https://github.com/qinenergy/cotta
[13] https://github.com/BIT-DA/RoTTA
[14] https://github.com/Jhyun17/DeYO
[15] https://github.com/Gorilla-Lab-SCUT/TRIBE
[16] https://github.com/BlueWhaleLab/COME

overconfident errors. Its core idea is to replace standard softmax entropy with an entropy-of-opinion objective that allocates explicit uncertainty mass, preventing the model from becoming overly certain on out-of-distribution inputs. In the official implementation, this is realized by exponentiating the logits to obtain nonnegative evidence values, then normalizing each class evidence by the sum of evidences plus a fixed constant; the same denominator also implicitly assigns the remaining probability mass to an explicit uncertainty term (with the constant fixed in code). For optimization, we use SGD for both backbones: for ResNet-50 we set the learning rate to $2.5 \times 10^{-4}$, and for ViT-B/16 we set the learning rate to $1.0 \times 10^{-3}$. We follow the original COME procedure and hyperparameters from the official protocol.

**ViDA**[17] (**Liu et al., 2024**) separates the adaptation process into acquiring domain-specific and domain-shared knowledge by injecting adapters with different embedding dimensions into the pre-trained model. It utilizes a Homeostatic Knowledge Allotment (HKA) strategy to dynamically fuse outputs from these adapters based on prediction uncertainty derived from MC Dropout. The framework relies on a teacher-student consistency objective. For optimization, we use SGD for both backbones: for ResNet-50 we set the learning rate to $2.5 \times 10^{-4}$, and for ViT-B/16 we set the learning rate to $1.0 \times 10^{-3}$. Following the official configuration, we set the larger adapter dimension to 128, the smaller adapter dimension to 1, and the uncertainty threshold to 0.2.

**PTTA**[18] (**Ma et al., 2025**) maintains a memory bank of recent samples and uses a saliency-based indicator to identify potentially malicious instances in the incoming batch. It then purifies the adaptation signal by retrieving benign samples from memory and applying a mixup-based correction, adding a purification loss term to the base TTA objective. For optimization, we use SGD for both backbones: for ResNet-50 we set the learning rate to $2.5 \times 10^{-4}$, and for ViT-B/16 we set the learning rate to $1.0 \times 10^{-3}$. Following the PTTA protocol, we use a memory bank with maximum length 1000 and retrieve the top-1 benign sample; the mixup weight is set to $1/2$ accordingly. PTTA is implemented based on DeYO for comparisons. We follow the original PTTA procedure and hyperparameters from the official protocol.

**SPA**[19] (**Niu et al., 2025b**) improves robustness across diverse shifts by introducing a self-bootstrapping mechanism that stabilizes pseudo-labeling under aggressive test-time transformations. SPA augments inputs using a frequency-domain mask and noise injection to construct complementary views, and learns a small projection module to align representations under these views. For optimization, we use SGD for both backbones: for ResNet-50 we set the learning rate to $2.5 \times 10^{-4}$, and for ViT-B/16 we set the learning rate to $1.0 \times 10^{-3}$. In the ImageNet classification setting, the authors use mask ratio 0.2 and noise factor 0.4, and optimize the projector with learning rate 0.05 while updating normalization affine parameters with learning rate 0.01. We follow the original SPA procedure and hyperparameters from the official protocol.

**MGP** stabilizes TTA by projecting entropy gradients onto the orthogonal complement of a tracked low-rank subspace, blocking the leakage of spurious updates into protected knowledge directions. It maintains a layer-wise FIFO gradient buffer and applies Marchenko–Pastur bulk-edge separation to extract the dominant subspace, combined with inertial fusion for smooth Grassmann-manifold tracking. Following TENT (Wang et al., 2021), we adapt only the affine parameters of normalization layers. Consistent with (Niu et al., 2025b; Ma et al., 2025), MGP also enables the ETA-style (Niu et al., 2022) gate in Eq. 1 and consistency-based augmentation from (Cohen & Giryes, 2024) to further suppress the magnitude of raw pseudo-label noise before projection. For optimization, we use SGD with momentum 0.9: learning rate $2.5 \times 10^{-4}$ for ResNet-50 and $1.0 \times 10^{-3}$ for ViT-B/16. We set buffer size $n = 32$, maximum rank $r_{\max} = 32$, refresh interval $K = 100$, novelty threshold $\tau_{\text{novel}} = 0.75$, and batch size 64.

## A.4. Details of Residual Anisotropy Injection

For the stress test in Table 5, we perturb only the residual update $g_{t,\perp} \triangleq \mathbf{P}_t^\perp g_t^{\text{raw}}$ after projection but before the optimizer step. We sample a fixed random unit direction $u$ per parameter tensor and project it onto the residual space: $u \leftarrow (\mathbf{I} - \mathbf{P}_t)u/\|(\mathbf{I} - \mathbf{P}_t)u\|_2$. We apply rank-1 amplification along $u$:

$$g_{t,\perp}^{(\kappa)} \triangleq (\mathbf{I} + \kappa u u^\top)g_{t,\perp}. \tag{14}$$

For $g_{t,\perp} = \alpha u + g_{\perp\perp}$ with $u^\top g_{\perp\perp} = 0$, this yields $(1 + \kappa)$-fold gain along $u$; hence $\kappa = 0.5$ and 100 correspond to $1.5\times$ and $101\times$ amplification, respectively.

We report $\nu_t \triangleq \text{Std}(\{\lambda_i^\perp\})/\text{Mean}(\{\lambda_i^\perp\})$, where $\{\lambda_i^\perp\}$ are eigenvalues of the residual-gradient covariance estimated from

---

[17] https://github.com/Yangsenqiao/vida
[18] https://github.com/HAIV-Lab/ptta
[19] https://github.com/mr-eggplant/SPA

a FIFO buffer (row-centered Gram matrix $\mathbf{C} = \frac{1}{n}\mathbf{G}\mathbf{G}^\top$), averaged over tracked parameters. This controlled violation increases anisotropy within $\mathcal{S}_t^\perp$ without introducing protected-subspace leakage (Lemma 4.1); hence $\nu$ can grow sharply while accuracy degradation remains moderate.

## B. Extended Experimental Results

### B.1. Two-Stage Abrupt Shifts and Absence of Adaptation Saturation

Since MGP restricts adaptation to the orthogonal residual space $\mathcal{S}_{\text{know}}^\perp$, a natural question is whether this residual capacity could eventually "saturate" or hinder adaptation when core semantics genuinely need to be revised. To stress-test this, we evaluate a Two-Stage Abrupt Shift on ViT-B/16:

- **Stage 1:** 20 rounds of adaptation on ImageNet-C (15 synthetic corruptions).

- **Stage 2:** Abrupt switch to a completely different ImageNet-K (Sketch) stream, with an additional imbalanced label distribution shift simulated by a Dirichlet distribution ($\beta = 0.1$) for another 20 rounds.

As shown in Table 6, most baselines either collapse during Stage 1 or fail to recover upon the abrupt shift in Stage 2. MGP, however, continues to improve in Stage 1 ($53.25 \rightarrow 55.69$) and successfully adapts further in Stage 2 ($36.68 \rightarrow 44.36$). This confirms that the residual space provides ample capacity for continuous learning without gradient starvation.

*Table 6.* **Two-Stage Abrupt Shift on ViT-B/16.** Accuracy (%) under consecutive 20-round adaptation on IN-C followed immediately by 20 rounds on imbalanced IN-K.

| Method | Stage 1: IN-C | | Stage 2: Imbalanced IN-K | |
|--------|---------|----------|---------|----------|
| | Round 1 | Round 20 | Round 1 | Round 20 |
| TENT | 36.89 | 0.10 | 0.24 | 0.21 |
| ETA | 49.61 | 0.11 | 0.44 | 0.42 |
| CoTTA | 40.34 | 41.31 | 30.40 | 30.48 |
| DeYO | 14.29 | 0.16 | 0.11 | 0.11 |
| PTTA | 24.67 | 0.12 | 0.05 | 0.04 |
| SPA | 49.56 | 0.09 | 0.13 | 0.13 |
| **Ours** | **53.25** | **55.69** | **36.68** | **44.36** |

### B.2. Robustness to Shuffled Domain Orders

To verify that the positive adaptation gains ($\Delta > 0$) observed in our main experiments are not merely an artifact of exploiting a fixed cyclic domain order, we evaluate MGP on a fully randomized (shuffled) stream. For ImageNet-C (ViT-B/16), domains are randomly permuted at the beginning of every round.

Table 7 shows that MGP remains perfectly stable and continues to accumulate beneficial updates ($53.13 \rightarrow 55.26$), confirming that the tracker dynamically provides a useful protected subspace even under rapidly evolving and unpredictable domain transitions.

*Table 7.* **Robustness to Shuffled Streams on ImageNet-C (ViT-B/16).** Domains are randomly permuted every round.

| Metric | TENT | ETA | CoTTA | SAR | DeYO | COME | SPA | PTTA | **Ours** |
|--------|------|-----|-------|-----|------|------|-----|------|----------|
| Acc. (R1) | 35.80 | 49.62 | 40.42 | 48.33 | 10.29 | 45.06 | 50.06 | 2.92 | **53.13** |
| Acc. (R40) | 0.10 | 0.11 | 40.37 | 47.29 | 0.16 | 0.12 | 0.14 | 0.13 | **55.26** |
| $\Delta$ | -35.70 | -49.51 | -0.05 | -1.04 | -10.13 | -44.94 | -49.92 | -2.79 | +2.13 |

## C. Additional Experimental Results

### C.1. Effect of Entropy Gating

To further investigate the limitations of sample-filtering approaches, we conduct experiments using TENT (Wang et al., 2021) augmented with entropy-based gating at various thresholds. Specifically, we filter out samples whose prediction entropy exceeds a predefined threshold before performing the update. Following the implementation in (Niu et al., 2022; 2023), the effective entropy threshold is computed as $\text{Thr} \times \ln(C)$, where $C$ is the number of classes (e.g., $C = 1000$ for ImageNet) and "Thr" denotes the relative coefficient reported in our tables. Lower thresholds correspond to stricter filtering (only samples with lower entropy are used for adaptation), while higher thresholds allow more samples to contribute.

Tables 8a and 8b report the average accuracy (%) across long-horizon cyclic revisits on ImageNet-C and ImageNet-R, respectively. The most restrictive threshold (0.10) delays collapse (11 rounds on ImageNet-C, 8 on ImageNet-R) but does not prevent it, indicating that even highly confident predictions carry harmful gradients. Looser thresholds ($\geq 0.40$) accelerate collapse to within 3 rounds. Crucially, no threshold achieves long-horizon stability, highlighting the fundamental limitation of sample-filtering: it controls *which* samples update but not the update *direction*. These results corroborate our hypothesis that collapse stems from directional leakage into $\mathcal{S}_{\text{know}}$, which entropy gating cannot address.

*Table 8.* **Effect of Entropy Gating Thresholds.** We vary the entropy gating threshold applied to TENT and report average accuracy (%) at each round on ImageNet-C/-R (ViT-B/16) under long-horizon cyclic revisits. We stop reporting once collapse occurs (accuracy $< 1\%$).

*(a)* ImageNet-C

| Thr. | R1 | R2 | R3 | R4 | R5 | R6 | R7 | R8 | R9 | R10 | R11 |
|------|-----|-----|-----|-----|-----|-----|-----|-----|-----|-----|-----|
| 0.1 | 46.45 | 49.47 | 49.97 | 49.99 | 49.70 | 49.46 | 49.52 | 49.18 | 43.87 | 1.81 | 0.10 |
| 0.2 | 47.64 | 43.26 | 0.18 | – | – | – | – | – | – | – | – |
| 0.4 | 37.46 | 0.15 | – | – | – | – | – | – | – | – | – |
| 0.6 | 37.60 | 0.08 | – | – | – | – | – | – | – | – | – |
| 0.8 | 36.99 | 0.15 | – | – | – | – | – | – | – | – | – |
| 1.0 | 37.31 | 0.14 | – | – | – | – | – | – | – | – | – |

*(b)* ImageNet-R

| Thr. | R1 | R2 | R3 | R4 | R5 | R6 | R7 | R8 |
|------|-----|-----|-----|-----|-----|-----|-----|-----|
| 0.1 | 49.01 | 54.15 | 56.02 | 56.89 | 57.17 | 57.14 | 24.93 | 0.50 |
| 0.2 | 51.11 | 55.66 | 14.15 | 0.48 | – | – | – | – |
| 0.4 | 40.93 | 0.62 | – | – | – | – | – | – |
| 0.6 | 23.35 | 0.49 | – | – | – | – | – | – |
| 0.8 | 22.40 | 0.49 | – | – | – | – | – | – |
| 1.0 | 22.40 | 0.49 | – | – | – | – | – | – |

### C.2. Extended Spectral Analysis

In the main text, we posit that the low-rank structure of reliable gradients is a common geometric property of test-time adaptation, rather than an artifact of a specific model or corruption type. Here, we provide empirical evidence supporting this claim across different architectures (CNNs vs. Transformers) and distribution shifts (Synthetic vs. Natural).

Fig. 8 presents the cumulative explained-variance curves of gradient second-moment matrices. Top Row: fixes architecture (ResNet-50) and varies datasets (Synthetic, Natural, and Geometric shifts). Bottom Row: fixes dataset (ImageNet-C) and varies model backbones, ranging from ResNets of increasing depth (18, 50, 101) to Transformer-based architectures (ViT, Swin). Across the settings, gradients from reliable samples (ER, Blue) exhibit a sharp "knee," reaching 90% variance within the top 10–20 dimensions, whereas spurious gradients (ES, Red) show a nearly linear, high-rank profile.

As shown in Fig. 8, the spectral separation between trustworthy adaptation signals and spurious noise appears robust across varying architectures, depths, and domains. This low-rank structure persists even under the abstract geometric deformations of ImageNet-R and becomes markedly more pronounced in larger models. Specifically, deeper ResNets (e.g., ResNet-101) display a more compact reliable subspace compared to shallower ones (ResNet-18). Furthermore, Vision Transformers (ViT, Swin) exhibit an even stronger low-rank tendency than CNNs, likely due to the sparse nature of attention mechanisms. These results collectively suggest that tracking a low-rank subspace $\mathcal{S}_t$ is a consistently effective strategy for capturing knowledge geometry while filtering out high-rank noise, across various datasets and architectures.

### C.3. Extended Subspace-Tracking Misalignment Analysis

In Sec. 5.4 and Fig. 4(b), we demonstrated that the unsupervised tracked subspace $\mathcal{S}_t$ maintains a low misalignment ($\delta_t^{\text{know}}$) with the oracle knowledge protected subspace $\mathcal{S}_{\text{know}}$ on ImageNet-C (ResNet-50).

To verify that this geometric property is broadly observed, we extend this analysis to a comprehensive suite of distribution shifts and diverse model architectures. We track the misalignment metric:

$$\delta_t^{\text{know}} \triangleq \sin \theta_{\max}(\mathcal{S}_{\text{know}}, \mathcal{S}_t) = \|\mathbf{P}_{\mathcal{S}_{\text{know}}} \mathbf{P}_t^{\perp}\|_2, \tag{15}$$

over the long-horizon adaptation process.

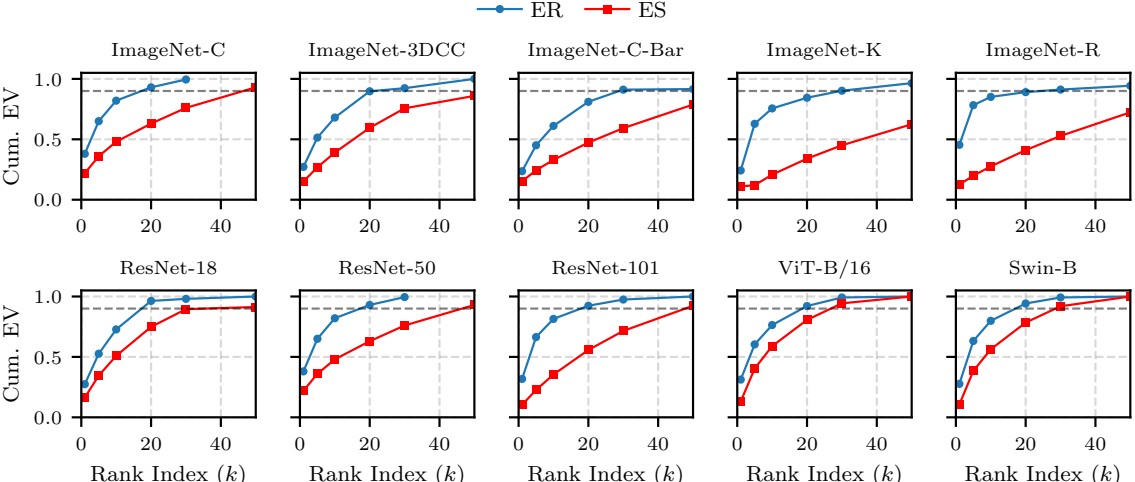

*Figure 8.* **Extended spectral analysis across architectures and datasets.** Cumulative Explained Variance (EV) of reliable (ER) vs. spurious (ES) gradients across varying settings. Top Row: Different domain shifts on ResNet-50. Bottom Row: Architecture comparison (Depth and Type). From left to right: ResNet-18, 50, 101, ViT-B/16, and Swin-B. In all cases, reliable adaptation signals reside in a low-dimensional subspace.

As shown in Fig. 9, MGP consistently maintains low misalignment across diverse datasets, and architectures (CNNs and Transformers). Whether facing synthetic corruptions (e.g., ImageNet-C, ImageNet-3DCC, ImageNet-C-Bar) or natural shifts (e.g., ImageNet-K, ImageNet-R), and regardless of model depth or type (e.g., ResNet-101, ViT-B/16, and Swin-B), the estimated subspace $\mathcal{S}_t$ closely tracks the oracle knowledge manifold $\mathcal{S}_{\text{know}}$. This empirically validates the assumption that $\mathcal{S}_{\text{know}}$ is spectrally dominant in the raw stream and can be robustly tracked without ground-truth labels, ensuring that the theoretical guarantees in Theorem 4.2 hold in broad scenarios.

## D. Proofs and Theoretical Details

**Proof Roadmap.** This section collects proofs for the theoretical statements in the main text and clarifies how they connect.

- **Geometry layer (misalignment).** Appendix D.1 defines the directional misalignment $\delta(\mathcal{A}, \mathcal{B}) = \|\mathbf{P}_{\mathcal{A}} \mathbf{P}_{\mathcal{B}}^{\perp}\|_2$ and relates it to principal angles (Appendix Lemma D.2).

- **Optimization layer (drift).** Appendix D.2 proves Proposition 3.3 (biased leakage yields linear expected drift). Appendix D.4 proves the deterministic reference-subspace drift bound (Theorem 4.2). Appendix D.5 gives a high-probability $\tilde{O}(\sqrt{T})$ bound under a martingale-difference (no coherent accumulation) condition (Assumption D.7).

- **Statistics layer (tracking).** Appendix D.7 provides a modular route from covariance estimation error to misalignment via Weyl + Davis–Kahan. Appendix D.8 closes the loop for our implementation: MP spike extraction with an *estimated* bulk edge (Eq. 5) plus Davis–Kahan yields misalignment control (Theorem D.23).

- **Behavioral connection.** Appendix D.9 shows that drift inside a protected subspace controls the output change attributable to those directions under a Jacobian bound.

### D.1. Preliminaries: projectors and principal angles

**Projectors and norms.** For any subspace $\mathcal{S} \subset \mathbb{R}^d$, let $\mathbf{P}_{\mathcal{S}}$ denote the orthogonal projector onto $\mathcal{S}$. If $\mathbf{U} \in \mathbb{R}^{d \times r}$ is an orthonormal basis of $\mathcal{S}$, then $\mathbf{P}_{\mathcal{S}} = \mathbf{U}\mathbf{U}^{\top}$ and $\mathbf{P}_{\mathcal{S}}^{\perp} = \mathbf{I} - \mathbf{P}_{\mathcal{S}}$. We use $\|\cdot\|_2$ for both Euclidean vector norm and matrix operator norm.

**Lemma D.1** (Basic properties of orthogonal projectors)**.** *Let $\mathbf{P}$ be an orthogonal projector and $\mathbf{P}^{\perp} = \mathbf{I} - \mathbf{P}$. Then:*

1. $\mathbf{P}\mathbf{P}^{\perp} = \mathbf{P}^{\perp}\mathbf{P} = \mathbf{0}$.
2. $\|\mathbf{P}\|_2 \le 1$ *and* $\|\mathbf{P}^{\perp}\|_2 \le 1$.

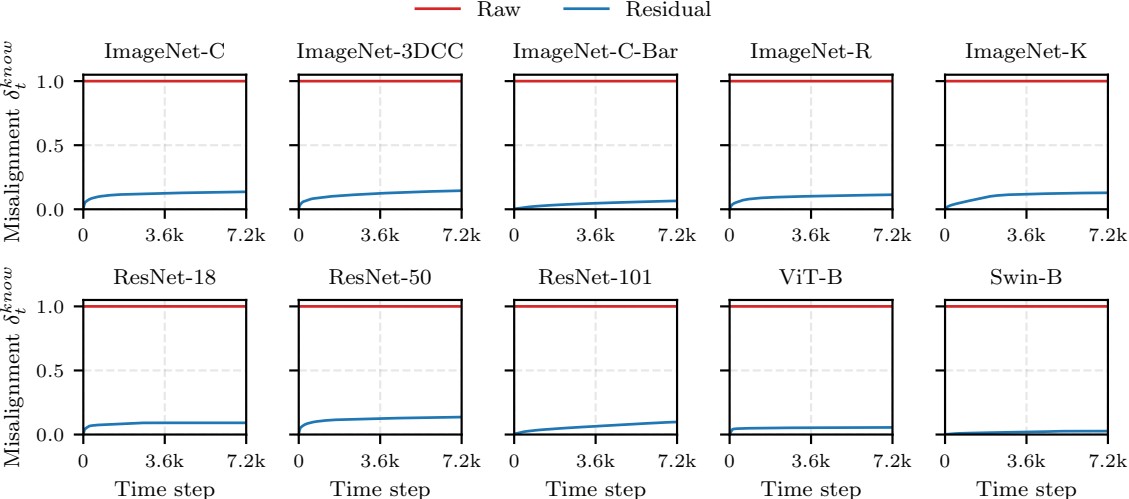

*Figure 9.* **Extended misalignment analysis.** Tracking the subspace misalignment $\delta_t^{\mathrm{know}}$ between the unsupervised tracked subspace $\mathcal{S}_t$ and the oracle knowledge protected subspace $\mathcal{S}_{\mathrm{know}}$. Top Row: Different domain shifts on ResNet-50. Bottom Row: Architecture comparison (Depth and Type). From left to right: ResNet-18, 50, 101, ViT-B/16, and Swin-B. In all cases, the tracked subspace (Blue) maintains consistently low misalignment compared to the baseline, confirming the robustness of spectral tracking across varying settings.

3. $\|\mathbf{A}\|_2 = \|\mathbf{A}^\top\|_2$ *for any matrix* $\mathbf{A}$.

*Proof.* (1) $\mathbf{P}\mathbf{P}^\perp = \mathbf{P}(\mathbf{I} - \mathbf{P}) = \mathbf{P} - \mathbf{P}^2 = \mathbf{0}$; similarly for $\mathbf{P}^\perp\mathbf{P}$. (2) $\mathbf{P}$ is symmetric idempotent so its eigenvalues lie in $\{0,1\}$ and $\|\mathbf{P}\|_2 \leq 1$; same for $\mathbf{P}^\perp$. (3) $\mathbf{A}$ and $\mathbf{A}^\top$ have identical singular values. □

**Principal angles and the "relative-to" convention.** Let $\mathcal{A}, \mathcal{B} \subset \mathbb{R}^d$ with $\dim(\mathcal{A}) = p$ and $\dim(\mathcal{B}) = q$. Let $\mathbf{U}_\mathcal{A} \in \mathbb{R}^{d \times p}$ and $\mathbf{U}_\mathcal{B} \in \mathbb{R}^{d \times q}$ be orthonormal bases. Define $k = \min\{p,q\}$ and principal angles $\theta_1 \leq \cdots \leq \theta_k$ by

$$\cos\theta_i = s_i\left(\mathbf{U}_\mathcal{A}^\top \mathbf{U}_\mathcal{B}\right), \quad i = 1, \ldots, k.$$

When $p > q$, $s_i(\cdot)$ denotes the $i$-th singular value, we extend the list to $p$ angles by setting $\theta_{q+1} = \cdots = \theta_p = \pi/2$. This makes the largest angle $\theta_{\max}(\mathcal{A}, \mathcal{B})$ well-defined for $\mathcal{A}$ *relative to* $\mathcal{B}$. Under this convention, if $p > q$ then necessarily $\theta_{\max}(\mathcal{A}, \mathcal{B}) = \pi/2$.

**Lemma D.2** (Directional misalignment equals sin of the largest principal angle). *Let* $\mathcal{A}, \mathcal{B} \subset \mathbb{R}^d$ *be subspaces with projectors* $\mathbf{P}_\mathcal{A}, \mathbf{P}_\mathcal{B}$. *Define*

$$\delta(\mathcal{A}, \mathcal{B}) \triangleq \|\mathbf{P}_\mathcal{A}\mathbf{P}_\mathcal{B}^\perp\|_2.$$

*Then*

$$\delta(\mathcal{A}, \mathcal{B}) = \sup_{\substack{u \in \mathcal{A} \\ \|u\|_2 = 1}} \|\mathbf{P}_\mathcal{B}^\perp u\|_2 = \sin\theta_{\max}(\mathcal{A}, \mathcal{B}) \in [0, 1],$$

*and* $\delta(\mathcal{A}, \mathcal{B}) = 0$ *iff* $\mathcal{A} \subseteq \mathcal{B}$.

*Proof.* By Lemma D.1(3),

$$\|\mathbf{P}_\mathcal{A}\mathbf{P}_\mathcal{B}^\perp\|_2 = \|(\mathbf{P}_\mathcal{A}\mathbf{P}_\mathcal{B}^\perp)^\top\|_2 = \|\mathbf{P}_\mathcal{B}^\perp\mathbf{P}_\mathcal{A}\|_2.$$

Thus

$$\|\mathbf{P}_\mathcal{B}^\perp\mathbf{P}_\mathcal{A}\|_2 = \sup_{\|x\|_2 = 1} \|\mathbf{P}_\mathcal{B}^\perp\mathbf{P}_\mathcal{A}x\|_2 = \sup_{\substack{u \in \mathcal{A} \\ \|u\|_2 = 1}} \|\mathbf{P}_\mathcal{B}^\perp u\|_2,$$

since $\mathbf{P}_\mathcal{A}x \in \mathcal{A}$ and every $u \in \mathcal{A}$ satisfies $u = \mathbf{P}_\mathcal{A}u$.

For unit $u \in \mathcal{A}$, $\|\mathbf{P}_\mathcal{B}^\perp u\|_2$ equals the sine of the angle between $u$ and $\mathcal{B}$. The variational characterization of principal angles (with the extension convention when $p > q$) gives the maximum such sine as $\sin\theta_{\max}(\mathcal{A}, \mathcal{B})$.

Finally, $\delta = 0$ iff $\mathbf{P}_\mathcal{B}^\perp u = 0$ for all $u \in \mathcal{A}$, i.e., $\mathcal{A} \subseteq \mathcal{B}$. □

**Lemma D.3** (Monotonicity of directional misalignment under superset protection). *Let $\mathcal{A}, \mathcal{B}_1, \mathcal{B}_2 \subset \mathbb{R}^d$ be subspaces with $\mathcal{B}_1 \subseteq \mathcal{B}_2$. Then*
$$\delta(\mathcal{A}, \mathcal{B}_2) \leq \delta(\mathcal{A}, \mathcal{B}_1).$$

*Proof.* Since $\mathcal{B}_1 \subseteq \mathcal{B}_2$, we have $\mathcal{B}_2^\perp \subseteq \mathcal{B}_1^\perp$, hence $\|\mathbf{P}_{\mathcal{B}_2}^\perp u\|_2 \leq \|\mathbf{P}_{\mathcal{B}_1}^\perp u\|_2$ for all $u$. Taking the supremum over unit $u \in \mathcal{A}$ and using Lemma D.2 yields the claim. $\square$

**Connection to the main text (direction-fixed).** Throughout the paper (and in particular in Theorem 4.2), we use
$$\delta_t \triangleq \|\mathbf{P}_\star \mathbf{P}_t^\perp\|_2 = \delta(\mathcal{S}_\star, \mathcal{S}_t) = \sin\theta_{\max}(\mathcal{S}_\star, \mathcal{S}_t).$$

### D.2. Proof of Proposition 3.3 (coherent biased leakage yields linear drift)

*Proof of Proposition 3.3.* Telescoping the gated update gives
$$\theta_T - \theta_0 = -\eta \sum_{t=0}^{T-1} \mathbb{I}_t g_t. \tag{16}$$

Projecting onto $\mathcal{S}_{\text{know}}$ yields the total protected drift
$$\Delta_T = -\eta \sum_{t=0}^{T-1} \mathbb{I}_t \mathbf{P}_{\text{know}} g_t. \tag{17}$$

Using the oracle bookkeeping split $g_t = g_t^{\text{ER}} + g_t^{\text{ES}}$, where $g_t^{\text{ES}} = \mathbb{J}_t g_t$ and $g_t^{\text{ER}} = (1 - \mathbb{J}_t) g_t$, we obtain
$$\Delta_T = -\eta \sum_{t=0}^{T-1} \mathbb{I}_t \mathbf{P}_{\text{know}} g_t^{\text{ER}} - \eta \sum_{t=0}^{T-1} \mathbb{I}_t \mathbf{P}_{\text{know}} g_t^{\text{ES}}. \tag{18}$$

The second term is, by definition, the ES-induced protected component
$$\Delta_T^{\text{ES}} \triangleq -\eta \sum_{t=0}^{T-1} \mathbb{I}_t \mathbf{P}_{\text{know}} g_t^{\text{ES}}. \tag{19}$$

Taking expectations and using linearity gives
$$\mathbb{E}[\Delta_T^{\text{ES}}] = -\eta \sum_{t=0}^{T-1} \mathbb{E}\left[\mathbb{I}_t \mathbf{P}_{\text{know}} g_t^{\text{ES}}\right] = -\eta \sum_{t=0}^{T-1} \mu_t^{\text{ES}}. \tag{20}$$

If $\mu_t^{\text{ES}} \equiv \mu^{\text{ES}} \neq 0$, then $\mathbb{E}[\Delta_T^{\text{ES}}] = -\eta T \mu^{\text{ES}}$. More generally, linear growth follows whenever $\left\|\sum_{t=0}^{T-1} \mu_t^{\text{ES}}\right\|_2 = \Omega(T)$. $\square$

---

*Remark* D.4 (Fully general identity (no stationarity)). Define $\mu_t \triangleq \mathbb{E}[\mathbb{I}_t \mathbf{P}_{\text{know}} g_t^{\text{ES}}]$. Then the identity $\mathbb{E}[\Delta_T] = -\eta \sum_{t=0}^{T-1} \mu_t$ always holds. Linear growth follows whenever $\left\|\sum_{t=0}^{T-1} \mu_t\right\|_2 = \Omega(T)$.

---

### D.3. Proof of Lemma 4.1 (exact zero drift in the tracked subspace)

*Proof of Lemma 4.1.* Under the MGP update in Eq. (4),
$$\theta_{t+1} - \theta_t = -\eta \mathbf{P}_t^\perp g_t^{\text{raw}}.$$

Left-multiplying by $\mathbf{P}_t$ and using Lemma D.1(1),
$$\mathbf{P}_t(\theta_{t+1} - \theta_t) = -\eta \mathbf{P}_t \mathbf{P}_t^\perp g_t^{\text{raw}} = \mathbf{0}.$$

This holds for every $t$. $\square$

> *Remark* D.5 (Indexing consistency with Algorithm 1). Algorithm 1 uses the currently available projector $\mathbf{P}_t = \mathbf{U}_t \mathbf{U}_t^\top$ to project the step-$t$ gradient when updating from $\theta_t$ to $\theta_{t+1}$. For theoretical clarity, define $\mathbf{P}_t$ as the projector *available at the moment of updating from $\theta_t$ to $\theta_{t+1}$*, which matches the role of $\mathbf{U}_t \mathbf{U}_t^\top$ in the algorithm. Under this convention the algebra above is exact.

### D.4. Proof of Theorem 4.2 (reference-subspace drift bound)

*Proof of Theorem 4.2.* Start from the projected update $\theta_{t+1} = \theta_t - \eta \, \mathbf{P}_t^\perp g_t^{\mathrm{raw}}$. Telescoping gives

$$\theta_T - \theta_0 = -\eta \sum_{t=0}^{T-1} \mathbf{P}_t^\perp g_t^{\mathrm{raw}}.$$

Project onto $\mathcal{S}_\star$ with projector $\mathbf{P}_\star$:

$$\mathbf{P}_\star(\theta_T - \theta_0) = -\eta \sum_{t=0}^{T-1} \mathbf{P}_\star \mathbf{P}_t^\perp g_t^{\mathrm{raw}}.$$

Take norms and apply triangle inequality:

$$F_T(\mathcal{S}_\star) = \|\mathbf{P}_\star(\theta_T - \theta_0)\|_2 \leq \eta \sum_{t=0}^{T-1} \|\mathbf{P}_\star \mathbf{P}_t^\perp g_t^{\mathrm{raw}}\|_2.$$

For each term, $\|\mathbf{A}x\|_2 \leq \|\mathbf{A}\|_2 \|x\|_2$ yields

$$\|\mathbf{P}_\star \mathbf{P}_t^\perp g_t^{\mathrm{raw}}\|_2 \leq \|\mathbf{P}_\star \mathbf{P}_t^\perp\|_2 \|g_t^{\mathrm{raw}}\|_2 = \delta_t \|g_t^{\mathrm{raw}}\|_2.$$

Summing completes the proof. □

**Corollary D.6** (Useful relaxations). *Under the conditions of Theorem 4.2:*

1. *(Cauchy–Schwarz)*

$$F_T(\mathcal{S}_\star) \leq \eta \Big(\sum_{t=0}^{T-1} \delta_t^2\Big)^{1/2} \Big(\sum_{t=0}^{T-1} \|g_t^{\mathrm{raw}}\|_2^2\Big)^{1/2}.$$

2. *If $\delta_t \leq \bar{\delta}$ and $\|g_t^{\mathrm{raw}}\|_2 \leq G$ for all $t$, then*

$$F_T(\mathcal{S}_\star) \leq \eta \, \bar{\delta} \, G \, T.$$

*Proof.* (1) Apply Cauchy–Schwarz to $\sum_t \delta_t \|g_t^{\mathrm{raw}}\|_2$. (2) Substitute the uniform bounds into Theorem 4.2. □

### D.5. From misalignment to *sublinear* protected drift: a martingale bound

Theorem 4.2 is deterministic and can be loose over long horizons since it upper bounds $\big\|\sum_t \mathbf{P}_\star \mathbf{P}_t^\perp g_t^{\mathrm{raw}}\big\|_2$ by a sum of magnitudes. Empirically, residual updates in $\mathcal{S}_t^\perp$ often do not accumulate coherently in protected directions. We formalize this by a martingale-difference (no coherent accumulation) condition.

**Assumption D.7** (No coherent accumulation in protected directions (martingale difference)). Let $\mathcal{F}_t$ be the filtration generated by the stream and algorithm up to time $t$. Assume

$$\mathbb{E}\big[\mathbf{P}_\star \mathbf{P}_t^\perp g_t^{\mathrm{raw}} \mid \mathcal{F}_{t-1}\big] = \mathbf{0} \qquad \text{for all } t. \tag{21}$$

**Lemma D.8** (Hilbert-space Azuma/Pinelis inequality (dimension-free; predictable bounds)). *Let $(X_t)_{t=0}^{T-1}$ be a martingale-difference sequence in $\mathbb{R}^d$ w.r.t. $\mathcal{F}_t$, i.e., $\mathbb{E}[X_t \mid \mathcal{F}_{t-1}] = 0$. Assume $\|X_t\|_2 \leq b_t$ almost surely, where $b_t$ may be random but is $\mathcal{F}_{t-1}$-measurable (predictable). Then for any $\rho \in (0, 1)$, with probability at least $1 - \rho$,*

$$\left\|\sum_{t=0}^{T-1} X_t\right\|_2 \leq \sqrt{2 \log(1/\rho)} \left(\sum_{t=0}^{T-1} b_t^2\right)^{1/2}. \tag{22}$$

*Proof.* This is a standard concentration inequality for martingales in Hilbert spaces (dimension-free). One route is via exponential supermartingales for $\langle u, \sum_t X_t \rangle$ for fixed $\|u\|_2 \le 1$, followed by taking the supremum over the unit ball. $\qquad \square$

**Theorem D.9** (Sublinear protected drift under misalignment + martingale difference). *Assume Assumption D.7. If $\|g_t^{\mathrm{raw}}\|_2 \le G$ for all $t$, then for any $\rho \in (0,1)$, with probability at least $1 - \rho$,*

$$F_T(\mathcal{S}_\star) = \|\mathbf{P}_\star(\theta_T - \theta_0)\|_2 \le \eta \, G \, \sqrt{2 \log(1/\rho)} \left( \sum_{t=0}^{T-1} \delta_t^2 \right)^{1/2} \le \eta \, G \left( \sup_{t<T} \delta_t \right) \sqrt{2T \log(1/\rho)}. \tag{23}$$

*Proof.* By telescoping under MGP updates,

$$\mathbf{P}_\star(\theta_T - \theta_0) = -\eta \sum_{t=0}^{T-1} \mathbf{P}_\star \mathbf{P}_t^{\perp} g_t^{\mathrm{raw}}.$$

Let $X_t \triangleq \mathbf{P}_\star \mathbf{P}_t^{\perp} g_t^{\mathrm{raw}}$. Assumption D.7 gives $\mathbb{E}[X_t \mid \mathcal{F}_{t-1}] = 0$. Moreover,

$$\|X_t\|_2 \le \|\mathbf{P}_\star \mathbf{P}_t^{\perp}\|_2 \, \|g_t^{\mathrm{raw}}\|_2 \le \delta_t \, G.$$

Since $\delta_t$ is determined by $\mathbf{P}_t$ available before drawing $g_t^{\mathrm{raw}}$ at step $t$, the bound $b_t = \delta_t G$ is predictable. Apply Lemma D.8 and multiply by $\eta$. $\qquad \square$

### D.6. Why small leakage can dominate over long horizons: bias vs. variance

**Proposition D.10** (Biased leakage yields $\Theta(T)$ mean drift; stochastic deviation is $O(\sqrt{T})$ in RMS). *Let $\ell_t \in \mathbb{R}^d$ be leakage vectors inside a fixed subspace (e.g., $\ell_t = \mathbf{P}_{\mathrm{know}} g_t^{\mathrm{ES}}$), and define the cumulative leakage drift $\Delta_T = -\eta \sum_{t=0}^{T-1} \ell_t$. Assume $\{\ell_t\}$ are independent with $\mathbb{E}[\ell_t] = \mu$ and $\mathbb{E}\|\ell_t - \mu\|_2^2 \le \sigma_\ell^2$. Then*

$$\mathbb{E}[\Delta_T] = -\eta \, T \, \mu, \qquad \mathbb{E}\|\Delta_T - \mathbb{E}\Delta_T\|_2^2 \le \eta^2 \, T \, \sigma_\ell^2.$$

*In particular,*

$$\|\mathbb{E}[\Delta_T]\|_2 = \Theta(\eta T \|\mu\|_2), \quad while \quad \sqrt{\mathbb{E}\|\Delta_T - \mathbb{E}\Delta_T\|_2^2} = O(\eta \sqrt{T} \, \sigma_\ell).$$

*Hence any nonzero bias $\mu$ dominates the stochastic term after $T \gtrsim \sigma_\ell^2 / \|\mu\|_2^2$.*

*Proof.* Linearity gives $\mathbb{E}[\Delta_T] = -\eta \sum_{t=0}^{T-1} \mathbb{E}[\ell_t] = -\eta T \mu$. Let $z_t = \ell_t - \mu$, so $\mathbb{E}[z_t] = 0$ and $\mathbb{E}\|z_t\|_2^2 \le \sigma_\ell^2$. Then $\Delta_T - \mathbb{E}\Delta_T = -\eta \sum_{t=0}^{T-1} z_t$ and by independence,

$$\mathbb{E}\left\| \sum_{t=0}^{T-1} z_t \right\|_2^2 = \sum_{t=0}^{T-1} \mathbb{E}\|z_t\|_2^2 \le T \sigma_\ell^2.$$

Multiply by $\eta^2$. $\qquad \square$

**Corollary D.11** (A weak high-probability deviation bound (Chebyshev)). *Under Proposition D.10, for any $\rho \in (0,1)$,*

$$\mathbb{P}\left( \|\Delta_T - \mathbb{E}\Delta_T\|_2 \ge \eta \, \sigma_\ell \, \sqrt{T/\rho} \right) \le \rho.$$

*Proof.* Apply Markov/Chebyshev to $\|\Delta_T - \mathbb{E}\Delta_T\|_2^2$ using Proposition D.10. $\qquad \square$

---

*Remark* D.12 (Dependence and martingale extensions). Independence is used only for the second-moment calculation. Analogous $O(\sqrt{T})$ RMS (and sharper concentration) results hold under standard martingale-difference or mixing assumptions.

---

## D.7. Subspace Estimation Error $\Rightarrow$ Misalignment Control

This section provides a modular bridge from spectral subspace estimation to the misalignment quantity $\delta_t = \|\mathbf{P}_\star \mathbf{P}_t^\perp\|_2$ used in Theorem 4.2. The results are conditional: they apply whenever a buffer covariance estimator is accurate in operator norm.

**Sample covariance from a gradient buffer.** Let $\mathbf{G} \in \mathbb{R}^{n \times d}$ stack $n$ gradient samples (rows) collected in a buffer at some tracking time. Let $\bar{g} = \frac{1}{n} \sum_{i=1}^n g_i \in \mathbb{R}^d$ be the sample mean. Define the centered matrix $\mathbf{G}_{\mathrm{cen}} = \mathbf{G} - \mathbf{1}\bar{g}^\top$ and its covariance

$$\widehat{\mathbf{\Sigma}} \triangleq \frac{1}{n}\mathbf{G}_{\mathrm{cen}}^\top \mathbf{G}_{\mathrm{cen}}.$$

**Lemma D.13** (Row-centering is a rank-1 covariance correction)**.** *Let* $\widehat{\mathbf{\Sigma}}_{\mathrm{raw}} = \frac{1}{n}\mathbf{G}^\top\mathbf{G}$ *and* $\widehat{\mathbf{\Sigma}} = \frac{1}{n}\mathbf{G}_{\mathrm{cen}}^\top\mathbf{G}_{\mathrm{cen}}$*. Then*

$$\widehat{\mathbf{\Sigma}} = \widehat{\mathbf{\Sigma}}_{\mathrm{raw}} - \bar{g}\,\bar{g}^\top,$$

*hence* $\widehat{\mathbf{\Sigma}} - \widehat{\mathbf{\Sigma}}_{\mathrm{raw}}$ *is negative semidefinite with rank at most 1 and* $\|\widehat{\mathbf{\Sigma}} - \widehat{\mathbf{\Sigma}}_{\mathrm{raw}}\|_2 = \|\bar{g}\|_2^2$.

*Proof.* Expand:

$$\mathbf{G}_{\mathrm{cen}}^\top \mathbf{G}_{\mathrm{cen}} = (\mathbf{G}^\top - \bar{g}\,\mathbf{1}^\top)(\mathbf{G} - \mathbf{1}\bar{g}^\top) = \mathbf{G}^\top\mathbf{G} - n\bar{g}\,\bar{g}^\top,$$

since $\mathbf{G}^\top\mathbf{1} = n\bar{g}$ and $\mathbf{1}^\top\mathbf{G} = n\bar{g}^\top$. Divide by $n$. $\qquad\square$

**A local spiked-covariance reference model.**

**Assumption D.14** (Local spiked covariance with eigengap)**.** There exists a fixed reference subspace $\mathcal{S}_\star = \mathrm{span}(\mathbf{U}_\star)$ with $\mathbf{U}_\star \in \mathbb{R}^{d \times r_\star}$ orthonormal, and a population covariance

$$\mathbf{\Sigma} = \mathbf{U}_\star \mathbf{\Lambda}_\star \mathbf{U}_\star^\top + \sigma^2 \mathbf{I},$$

where $\mathbf{\Lambda}_\star = \mathrm{diag}(\lambda_1^\star, \ldots, \lambda_{r_\star}^\star)$ with $\lambda_1^\star \geq \cdots \geq \lambda_{r_\star}^\star > 0$. Let $\lambda_1 \geq \cdots \geq \lambda_d$ be the eigenvalues of $\mathbf{\Sigma}$. Assume a strict eigengap at rank $r_\star$:

$$\mathrm{gap} \triangleq \lambda_{r_\star} - \lambda_{r_\star+1} > 0.$$

**Davis–Kahan (gap form) to control $\delta_t$.** Let $\widehat{\mathbf{U}} \in \mathbb{R}^{d \times r_\star}$ be the top-$r_\star$ eigenvectors of $\widehat{\mathbf{\Sigma}}$ and $\widehat{\mathbf{P}} = \widehat{\mathbf{U}}\widehat{\mathbf{U}}^\top$. Let $\mathbf{P}_\star = \mathbf{U}_\star\mathbf{U}_\star^\top$.

**Theorem D.15** (Subspace perturbation implies misalignment control (Davis–Kahan, operator form))**.** *Under Assumption D.14, let* $\widehat{\mathbf{\Sigma}} = \mathbf{\Sigma} + \mathbf{E}$*. If* $\|\mathbf{E}\|_2 \leq \mathrm{gap}/2$*, then*

$$\delta(\mathcal{S}_\star, \widehat{\mathcal{S}}) = \|\mathbf{P}_\star\widehat{\mathbf{P}}^\perp\|_2 = \|\sin\Theta(\mathcal{S}_\star, \widehat{\mathcal{S}})\|_2 \leq \frac{2\|\mathbf{E}\|_2}{\mathrm{gap}},$$

*where* $\widehat{\mathcal{S}} = \mathrm{span}(\widehat{\mathbf{U}})$.

*Proof.* Let $\mathbf{U}_\perp$ be an orthonormal basis of $\mathcal{S}_\star^\perp$. A standard identity gives $\|\sin\Theta(\mathcal{S}_\star, \widehat{\mathcal{S}})\|_2 = \|\mathbf{U}_\perp^\top\widehat{\mathbf{U}}\|_2$. Write the eigendecomposition split at $r_\star$: $\mathbf{\Sigma} = \mathbf{U}_\star\mathbf{\Lambda}\mathbf{U}_\star^\top + \mathbf{U}_\perp\mathbf{\Lambda}_\perp\mathbf{U}_\perp^\top$. Using $\widehat{\mathbf{\Sigma}}\widehat{\mathbf{U}} = \widehat{\mathbf{U}}\widehat{\mathbf{\Lambda}}$ and left-multiplying by $\mathbf{U}_\perp^\top$ yields

$$(\mathbf{\Lambda}_\perp - \widehat{\mathbf{\Lambda}})\,(\mathbf{U}_\perp^\top\widehat{\mathbf{U}}) = -\mathbf{U}_\perp^\top\mathbf{E}\widehat{\mathbf{U}}.$$

Taking operator norms, $\|\mathbf{U}_\perp^\top\widehat{\mathbf{U}}\|_2 \leq \|(\mathbf{\Lambda}_\perp - \widehat{\mathbf{\Lambda}})^{-1}\|_2 \|\mathbf{E}\|_2$. By Weyl, when $\|\mathbf{E}\|_2 \leq \mathrm{gap}/2$ the spectral separation is at least $\mathrm{gap}/2$, so $\|(\mathbf{\Lambda}_\perp - \widehat{\mathbf{\Lambda}})^{-1}\|_2 \leq 2/\mathrm{gap}$, proving the claim. $\qquad\square$

**Corollary D.16** (A concrete route to small $\delta_t$)**.** *If at a refresh time the covariance estimator satisfies* $\|\widehat{\mathbf{\Sigma}} - \mathbf{\Sigma}\|_2 \leq \varepsilon$ *and* $\varepsilon \leq \mathrm{gap}/2$*, then the resulting subspace satisfies*

$$\delta(\mathcal{S}_\star, \widehat{\mathcal{S}}) \leq \frac{2\varepsilon}{\mathrm{gap}}.$$

**Thresholded rank selection is stable under perturbations (strict-threshold form).**

**Lemma D.17** (Rank selection stability under a spectral threshold (Weyl; strict form)). *Let $\Sigma$ be symmetric with eigenvalues $\lambda_1 \geq \cdots \geq \lambda_d$ and let $\widehat{\Sigma} = \Sigma + \mathbf{E}$ with $\|\mathbf{E}\|_2 \leq \varepsilon$. Fix a threshold $\tau \in \mathbb{R}$. If for some integer $r$,*

$$\lambda_r > \tau + \varepsilon \quad and \quad \lambda_{r+1} < \tau - \varepsilon,$$

*then the number of eigenvalues of $\widehat{\Sigma}$ strictly above $\tau$ equals $r$:*

$$\left|\{i : \widehat{\lambda}_i > \tau\}\right| = r.$$

*Proof.* By Weyl's inequality, $|\widehat{\lambda}_i - \lambda_i| \leq \|\mathbf{E}\|_2 \leq \varepsilon$ for all $i$. Hence $\widehat{\lambda}_r \geq \lambda_r - \varepsilon > \tau$ and $\widehat{\lambda}_{r+1} \leq \lambda_{r+1} + \varepsilon < \tau$. $\square$

> *Remark* D.18 (How this closes the loop). Lemma D.17 and Theorem D.15 provide a clean conditional loop: if eigenvalues are separated around a threshold $\tau$ by a margin exceeding the covariance perturbation, then thresholding recovers the correct rank; and if the eigengap is large compared to the perturbation, then the recovered subspace has small misalignment.

### D.8. MP Spike Extraction $\Rightarrow$ Misalignment Control

We now specialize the generic thresholding + Davis–Kahan bridge to the MP bulk-edge spike extraction used by MGP. A key implementation detail is that the algorithm uses an *estimated* bulk edge $\tau_{\text{edge}}$ (Eq. 5), not an oracle edge.

**Assumption D.19** (Local spiked model with MP-separated bulk/edge margin). At a refresh time, the centered buffer gradients follow a local spiked model as in Assumption D.14, and let $\gamma = d/n$. Define the (oracle) MP bulk edge

$$\tau_{\text{MP}} \triangleq \sigma^2(1 + \sqrt{\gamma})^2.$$

Assume a margin $\Delta_{\text{gap}} > 0$ such that

$$\lambda_{r_\star}(\Sigma) \geq \tau_{\text{MP}} + \Delta_{\text{gap}}, \qquad \lambda_{r_\star+1}(\Sigma) \leq \tau_{\text{MP}} - \Delta_{\text{gap}}.$$

**Assumption D.20** (Edge estimation accuracy (algorithmic threshold near oracle edge)). Let $\tau_{\text{edge}}$ be the threshold used by the algorithm (Eq. 5). Assume

$$|\tau_{\text{edge}} - \tau_{\text{MP}}| \leq \varepsilon_\tau$$

for some $\varepsilon_\tau \geq 0$ at the refresh time.

> *Remark* D.21 (When is Assumption D.20 reasonable?). Median-based variance estimation is scale-equivariant and robust provided fewer than $50\%$ of the nonzero eigenvalues are spikes (i.e., $r_\star \ll n$), so that the sample median remains in the bulk. In that regime, $\tau_{\text{edge}}$ concentrates near $\tau_{\text{MP}}$, yielding a small $\varepsilon_\tau$ in practice.

**Lemma D.22** (MP threshold recovers the correct rank using the *estimated* edge). *Under Assumption D.19 and Assumption D.20, let $\widehat{\Sigma} = \Sigma + \mathbf{E}$ with $\|\mathbf{E}\|_2 \leq \varepsilon$. If*

$$\Delta_{\text{gap}} > \varepsilon_\tau + \varepsilon,$$

*then thresholding $\widehat{\Sigma}$ at $\tau_{\text{edge}}$ selects exactly $r_\star$ eigenvalues: $\left|\{i : \widehat{\lambda}_i > \tau_{\text{edge}}\}\right| = r_\star$.*

*Proof.* From Assumption D.20, $\tau_{\text{edge}} \leq \tau_{\text{MP}} + \varepsilon_\tau$ and $\tau_{\text{edge}} \geq \tau_{\text{MP}} - \varepsilon_\tau$. Thus, using Assumption D.19,

$$\lambda_{r_\star}(\Sigma) \geq \tau_{\text{MP}} + \Delta_{\text{gap}} > \tau_{\text{MP}} + \varepsilon_\tau + \varepsilon \geq \tau_{\text{edge}} + \varepsilon,$$

and

$$\lambda_{r_\star+1}(\Sigma) \leq \tau_{\text{MP}} - \Delta_{\text{gap}} < \tau_{\text{MP}} - \varepsilon_\tau - \varepsilon \leq \tau_{\text{edge}} - \varepsilon.$$

Therefore Lemma D.17 applies with $\tau = \tau_{\text{edge}}$. $\square$

**Theorem D.23** (MP spike extraction (with estimated edge) yields small misalignment). *Under Assumption D.19 and Assumption D.20, let $\widehat{\mathcal{S}}$ be the subspace returned by thresholding $\widehat{\boldsymbol{\Sigma}}$ at $\tau_{\text{edge}}$ and taking the corresponding eigenvectors. If*

$$\|\widehat{\boldsymbol{\Sigma}} - \boldsymbol{\Sigma}\|_2 \leq \min\left\{\text{gap}/2,\ \Delta_{\text{gap}} - \varepsilon_\tau\right\},$$

*then* $\dim(\widehat{\mathcal{S}}) = r_\star$ *and*

$$\delta(\mathcal{S}_\star, \widehat{\mathcal{S}}) = \|\mathbf{P}_\star \widehat{\mathbf{P}}^\perp\|_2 \leq \frac{2\|\widehat{\boldsymbol{\Sigma}} - \boldsymbol{\Sigma}\|_2}{\text{gap}}.$$

*Proof.* Let $\mathbf{E} = \widehat{\boldsymbol{\Sigma}} - \boldsymbol{\Sigma}$ and $\varepsilon = \|\mathbf{E}\|_2$. The condition $\varepsilon \leq \Delta_{\text{gap}} - \varepsilon_\tau$ implies $\Delta_{\text{gap}} > \varepsilon_\tau + \varepsilon$, so Lemma D.22 yields $\dim(\widehat{\mathcal{S}}) = r_\star$. The condition $\varepsilon \leq \text{gap}/2$ allows applying Theorem D.15, giving the misalignment bound. ☐

> *Remark* D.24 (From MP candidate subspace to the fused tracked subspace used in updates). Theorem D.23 controls the misalignment of the *candidate* spike subspace returned by thresholding. In Algorithm 1, we apply INERTIALFUSION and truncation. Whenever the fused protected subspace $\mathcal{S}_t$ contains the candidate spike subspace (e.g., mild over-estimation, or truncation preserves spikes), Lemma D.3 implies
>
> $$\delta(\mathcal{S}_\star, \mathcal{S}_t) \leq \delta(\mathcal{S}_\star, \widehat{\mathcal{S}}),$$
>
> so the MP/Davis–Kahan misalignment guarantee transfers (and can only improve) for the actual projector used in updates.

**Proposition D.25** (Effect of rank misspecification). *Let $\mathcal{S}_\star$ be the $r_\star$-dimensional reference subspace and let $\mathcal{S}_t$ be the tracked subspace with $\dim(\mathcal{S}_t) = r_t$ (selected by Eq. (6), clipped to $[1, r_{\max}]$).*

*(Over-estimation). If $r_t \geq r_\star$ and $\mathcal{S}_\star \subseteq \mathcal{S}_t$, then $\delta(\mathcal{S}_\star, \mathcal{S}_t) = 0$ (Lemma D.2). Therefore $F_T(\mathcal{S}_\star) = 0$ by Theorem 4.2.*

*(Under-estimation). If $r_t < r_\star$, then under the principal-angle convention $\theta_{\max}(\mathcal{S}_\star, \mathcal{S}_t) = \pi/2$, hence $\delta(\mathcal{S}_\star, \mathcal{S}_t) = 1$. In this case, Theorem 4.2 reduces to the vacuous bound $F_T(\mathcal{S}_\star) \leq \eta \sum_t \|g_t^{\text{raw}}\|_2$. Exact protection still holds on $\mathcal{S}_t$ (Lemma 4.1); remaining drift is attributable to missed directions.*

**(Optional) One-shot covariance concentration.** To obtain explicit rates, one may assume a snapshot sub-Gaussian model for buffered gradients and apply standard matrix concentration to bound $\|\widehat{\boldsymbol{\Sigma}} - \boldsymbol{\Sigma}\|_2$.

**Assumption D.26** (Sub-Gaussian gradients (snapshot model)). At a fixed refresh time, the buffered gradients are independent with mean 0 and covariance $\boldsymbol{\Sigma}$. Moreover, there exists $\kappa > 0$ such that

$$\sup_{\|u\|_2=1} \|\langle u, g\rangle\|_{\psi_2} \leq \kappa.$$

**Theorem D.27** (Operator-norm covariance concentration (intrinsic dimension form)). *Under Assumption D.26, for any $\rho \in (0, 1)$, with probability at least $1 - \rho$,*

$$\left\|\widehat{\boldsymbol{\Sigma}} - \boldsymbol{\Sigma}\right\|_2 \leq c\kappa^2 \|\boldsymbol{\Sigma}\|_2 \left(\sqrt{\frac{r_{\text{eff}} + \log(2/\rho)}{n}} + \frac{r_{\text{eff}} + \log(2/\rho)}{n}\right), \tag{24}$$

*where $c > 0$ is an absolute constant and $r_{\text{eff}} \triangleq \text{tr}(\boldsymbol{\Sigma})/\|\boldsymbol{\Sigma}\|_2$.*

> *Remark* D.28 (Scope). Theorem D.27 is standard (e.g., via matrix Bernstein/Tropp-style bounds). We use it only to instantiate sufficient conditions for small $\delta_t$; dependence/mixing variants can be substituted if desired.

**Corollary D.29** (A concrete bound on misalignment at one refresh). *Under Assumption D.19, Assumption D.20 and Assumption D.26, if the RHS of (24) is at most $\min\{\Delta_{\text{gap}} - \varepsilon_\tau, \text{gap}/2\}$, then $\dim(\widehat{\mathcal{S}}) = r_\star$ and*

$$\delta(\mathcal{S}_\star, \widehat{\mathcal{S}}) \leq \frac{2}{\text{gap}} \left\|\widehat{\boldsymbol{\Sigma}} - \boldsymbol{\Sigma}\right\|_2.$$

**Corollary D.30** (Uniform control over a horizon via union bound). *Suppose subspace refresh happens every $K$ steps over a horizon $T$, so that the number of refreshes is $M = \lceil T/K \rceil$. Applying Corollary D.29 with failure probability $\zeta/M$ and union bounding, with probability at least $1 - \zeta$,*

$$\sup_{t < T} \delta_t \lesssim \frac{\kappa^2 \|\Sigma\|_2}{\text{gap}} \left( \sqrt{\frac{r_{\text{eff}} + \log(M/\zeta)}{n}} + \frac{r_{\text{eff}} + \log(M/\zeta)}{n} \right).$$

*Remark* D.31 (Closing the loop to drift). Combining Corollary D.30 with Theorem 4.2 yields an explicit tracking-limited bound on $F_T(\mathcal{S}_\star)$. Under Assumption D.7, Theorem D.9 further gives the high-probability sublinear variant.

## D.9. A lightweight behavioral connection: drift inside $\mathcal{S}_\star$ bounds functional change

**Assumption D.32** (Directional Lipschitz / Jacobian control along $\mathcal{S}_\star$). For any input $x$, define $\phi(\theta) = f_\theta(x)$ (e.g., logits or embeddings). Assume $\phi$ is differentiable along the adaptation path and there exists $L_\star(x) \geq 0$ such that

$$\sup_{\theta \text{ on path}} \|\nabla_\theta \phi(\theta) \mathbf{P}_\star\|_2 \leq L_\star(x).$$

**Lemma D.33** (Protected-direction drift controls the output change attributable to $\mathcal{S}_\star$). *Under Assumption D.32, let $\Delta_\star = \mathbf{P}_\star(\theta_T - \theta_0)$. Then*

$$\left\| f_{\theta_0 + \Delta_\star}(x) - f_{\theta_0}(x) \right\|_2 \leq L_\star(x) \|\Delta_\star\|_2 = L_\star(x) F_T(\mathcal{S}_\star).$$

*Proof.* Consider $\theta(\alpha) = \theta_0 + \alpha \Delta_\star$ for $\alpha \in [0, 1]$. By the fundamental theorem of calculus,

$$f_{\theta_0 + \Delta_\star}(x) - f_{\theta_0}(x) = \int_0^1 \nabla_\theta f_{\theta(\alpha)}(x) \, \Delta_\star \, d\alpha.$$

Since $\Delta_\star = \mathbf{P}_\star \Delta_\star$,

$$\|f_{\theta_0 + \Delta_\star}(x) - f_{\theta_0}(x)\|_2 \leq \int_0^1 \|\nabla_\theta f_{\theta(\alpha)}(x) \mathbf{P}_\star\|_2 \|\Delta_\star\|_2 \, d\alpha \leq L_\star(x) \|\Delta_\star\|_2.$$

$\square$

*Remark* D.34 (Interpretation for collapse avoidance). Lemma D.33 isolates the behavioral effect contributed purely by protected-direction drift. Combined with Theorem 4.2, it makes the following logic explicit: small misalignment $\delta_t \Rightarrow$ small $F_T(\mathcal{S}_\star) \Rightarrow$ limited functional drift attributable to protected directions.

# E. Random Matrix Theory for MP Bulk-Edge Spectral Distillation

This appendix justifies the MP-based bulk-edge thresholding used in Eq. (5)–(6). We emphasize that MP modeling is applied to the *approximately isotropic residual-like* component of gradients, motivated by empirical near-isotropy in $\mathcal{S}_{\text{know}}^\perp$.

## E.1. MP bulk edge for isotropic noise (nonzero eigenvalues, normalized)

**Setup (noise-only).** Let $\mathbf{Z} \in \mathbb{R}^{n \times d}$ have i.i.d. entries with mean 0, variance $\sigma^2$, and finite fourth moment. Consider the sample covariance

$$\mathbf{C} \triangleq \frac{1}{n} \mathbf{Z}^\top \mathbf{Z}.$$

Let $\lambda_1 \geq \cdots \geq \lambda_{\min(n,d)} > 0$ denote its *nonzero* eigenvalues. Equivalently, if $s_i(\mathbf{Z})$ are singular values then $\lambda_i = s_i(\mathbf{Z})^2/n$.

**Theorem E.1** (Marchenko–Pastur (MP) bulk support for nonzero eigenvalues; classical). *Let $\gamma = d/n$ converge to a positive constant as $n, d \to \infty$. Then the empirical distribution of the nonzero eigenvalues of $\mathbf{C} = \frac{1}{n} \mathbf{Z}^\top \mathbf{Z}$ converges almost surely to a deterministic law supported on*

$$[\lambda_-, \lambda_+], \qquad \lambda_\pm = \sigma^2 (1 \pm \sqrt{\gamma})^2.$$

*Moreover, the largest nonzero eigenvalue satisfies $\lambda_1 \to \lambda_+$ almost surely.*

*Remark* E.2 (Why we ignore the zero-eigenvalue atom when $d \gg n$). When $\gamma > 1$ (typical since $d \gg n$), $\mathbf{C}$ has $(d - n)$ exact zero eigenvalues. Our implementation computes only the $\min(n, d) = n$ nonzero eigenvalues via SVD of $\mathbf{G}_{\text{cen}}$, so all MP quantities here refer to the *nonzero-eigenvalue* MP density normalized to integrate to 1 over $[\lambda_-, \lambda_+]$.

### E.2. Median-based variance estimation (scale equivariance + robustness intuition)

**MP median definition (nonzero eigenvalues; normalized).** Let $m_\gamma$ denote the median of the MP *nonzero-eigenvalue* bulk when $\sigma^2 = 1$:

$$\int_{\lambda_-^{(1)}}^{m_\gamma} \rho_\gamma(\lambda)\, d\lambda = \frac{1}{2}, \qquad \lambda_\pm^{(1)} = \left(1 \pm \sqrt{\gamma}\right)^2,$$

where $\rho_\gamma$ is the MP density normalized so that $\int_{\lambda_-^{(1)}}^{\lambda_+^{(1)}} \rho_\gamma(\lambda)\, d\lambda = 1$.

**Lemma E.3** (Median is scale-equivariant). *Let $X \geq 0$ be a random variable with median $\mathrm{med}(X)$. For any scalar $a > 0$, $\mathrm{med}(aX) = a\,\mathrm{med}(X)$.*

*Proof.* Let $m = \mathrm{med}(X)$. Then $\mathbb{P}(X \leq m) \geq \frac{1}{2}$ and $\mathbb{P}(X \geq m) \geq \frac{1}{2}$. Hence $\mathbb{P}(aX \leq am) = \mathbb{P}(X \leq m) \geq \frac{1}{2}$ and similarly $\mathbb{P}(aX \geq am) \geq \frac{1}{2}$. $\qquad\square$

**Justification of Eq. (5).** Under MP scaling, bulk eigenvalues satisfy $\lambda \overset{d}{=} \sigma^2 Z$ where $Z$ follows the $\sigma^2 = 1$ MP bulk. By Lemma E.3,

$$\mathrm{med}(\lambda) = \sigma^2\, m_\gamma.$$

Rearranging yields the estimator

$$\hat{\sigma}^2 = \frac{\mathrm{median}(\{\lambda_i\})}{m_\gamma}, \qquad \tau_{\text{edge}} = \hat{\sigma}^2 \left(1 + \sqrt{\gamma}\right)^2,$$

which matches Eq. (5).

*Remark* E.4 (Robustness intuition for the median estimate). The median estimate is robust as long as fewer than $50\%$ of the computed nonzero eigenvalues are spikes, so the median remains in the bulk. This matches the intended regime $r \ll n$.

### E.3. Row-centering and its spectral impact

In practice we use row-centering $\mathbf{G}_{\text{cen}}$ before SVD. Centering modifies the raw covariance by a rank-1 correction (Lemma D.13), hence it cannot create spurious high-rank structure.

### E.4. Applicability to gradient buffers

The MP model is not assumed to hold for the full raw gradient stream. Rather, it is used as a robust bulk model for the approximately isotropic residual component (cf. Fig. 3d). The MP edge criterion is thus a conservative, stability-oriented spike extractor; when combined with perturbation arguments (Appendix D.8), it yields misalignment control whenever spikes are separated from the bulk edge by a margin exceeding covariance and threshold-estimation errors.

### E.5. How we compute $m_\gamma$

We compute $m_\gamma$ by solving for the root of

$$\phi(m) \triangleq \int_{\lambda_-^{(1)}}^{m} \rho_\gamma(\lambda)\, d\lambda - \frac{1}{2}$$

over $m \in [\lambda_-^{(1)}, \lambda_+^{(1)}]$ (monotone 1D bisection), and cache it per $\gamma = d/n$.

