# OpenReview forum: "Blocking the Leakage: Manifold-Aware Gradient Projection for Long-Horizon Test-Time Adaptation"
_ICML.cc/2026/Conference — ICML 2026 regular_

### Official Review · Reviewer_L1EY · 2026-03-10

**Soundness:** 3
**Presentation:** 4
**Significance:** 3
**Originality:** 3
**Overall Recommendation:** 6
**Confidence:** 4

**Summary:**

This work introduces MGP, a geometric framework supported by spectral analysis, to ensure the long-horizon stability of TTA, motivated by the hypothesis that pre-trained models are susceptible to manifold erosion. Extensive experiments across representative TTA benchmarks verify that MGP can effectively prevent collapse while preserving adaptation plasticity, offering a geometric solution to the instability of online entropy minimization.

**Compliance With Llm Reviewing Policy:**

Affirmed.

**Final Justification:**

This paper delivers valuable insights and is supported by thorough and convincing experiments. I believe it is a high-quality paper and deserves to be accepted.

**Key Questions For Authors:**

see weaknesses

**Limitations:**

yes

**Strengths And Weaknesses:**

**Strengths**
- The main innovation of this paper lies in its identification of knowledge-subspace erosion caused by spurious gradient directions, which cannot be effectively removed by simple magnitude-based filtering.
- The proposed method is simple, targeted, and well aligned with the underlying diagnosis.
- The paper is well structured overall, featuring a clear introduction and informative figures.
- The experiments are comprehensive and convincing, providing solid support for the paper’s claims and for the effectiveness of MGP.

**Weaknesses**
- MGP requires maintaining additional FIFO buffers, performing periodic refreshes, and applying non-negligible projection operations, which together introduce extra computational and memory overhead. While Table 1 summarizes the runtime comparison, memory consumption is not quantitatively compared.

- Since MGP strictly constrains updates to the residual space $S_t^\perp$ and preserves the protected subspace by design, it may hinder adaptation when the target shift genuinely requires modifying core semantic or discriminative directions rather than only exploiting residual degrees of freedom. While the paper argues that the residual space is sufficient for adaptation, it does not systematically evaluate settings where the core knowledge subspace itself must be revised.

- The experiments restrict updates to normalization layers only, thus the observed geometric behavior and gains of MGP may partially depend on this limited setup. It remains unclear whether the same advantages would hold under broader adaptation regimes, such as larger trainable subsets or full-parameter test-time adaptation.

- The method relies on the assumption that reliable gradients form a stable low-rank dominant subspace that can be tracked online from unlabeled raw gradients via spectral separation. While the paper includes stress tests on tracking lag and residual anisotropy, these perturbations are still relatively mild, leaving it unclear whether the tracker would remain reliable under more challenging conditions, such as very low ER proportions, weak source models, highly unfamiliar target domains, or rapidly evolving dominant directions.

---

> ### Author Rebuttal · Authors · 2026-03-30
>
> We appreciate your valuable feedback and constructive comments. We address your questions below.
>
> > Q1: Please quantify memory overhead, not only runtime
>
> **A1.** Thank you for your suggestion. In the default normalization-only setting, MGP's memory overhead comes from maintaining the buffer over the active trainable parameters (about 0.21% of the network), rather than over the full model. We now report peak GPU memory explicitly on ImageNet-C (IN-C) with ResNet-50:
>
> | Metric | Tent | ETA | CoTTA | RoTTA | SAR | DeYO | TRIBE | COME | PTTA | SPA | Ours |
> | --- | :---: | :---: | :---: | :---: | :---: | :---: | :---: | :---: | :---: | :---: | :---: |
> | Time / 1k imgs | 1.30 | 1.37 | 23.21 | 31.10 | 2.77 | 1.95 | 15.19 | 1.30 | 3.61 |8.84 | 4.28 |
> | Peak GPU Memory (GB) | 8.21 | 8.21 | 14.22 | 15.50 | 14.22 | 8.88 | 17.46 | 8.21 | 13.46 | 18.56 | 13.36 |
>
> > Q2: Could protecting the core subspace hinder adaptation when the target truly requires changing core semantics?
>
> **A2.** We agree that a strict barrier on $\mathcal{S}_{\mathrm{know}}$ may be conservative if core semantics need to be revised. However, our focus is long-horizon TTA under label-preserving shifts, where protecting this discriminative subspace is important. Empirically, we also ran a two-stage abrupt-shift test on ViT-B/16: after 20 rounds on IN-C, we switch to a substantially different ImageNet-K (IN-K) stream with an imbalanced label-distribution shift simulated by a Dirichlet distribution ($\beta = 0.1$) for another 20 rounds. Our method improves in Stage 1 (53.25 $\to$ 55.69) and continues adapting in Stage 2 (36.68 $\to$ 44.36). These results suggest that protecting the core subspace does not prevent meaningful adaptation within the horizons we study. Detailed results are in our response to Reviewer c9bv's Q2.
>
> > Q3: Would the same advantage hold beyond normalization-only adaptation?
>
> **A3.** Although the current paper focuses on the standard normalization-only TTA regime for the most thorough validation, MGP is not inherently restricted to normalization parameters. In additional broader-subset experiments on IN-C, its stability advantage remains clear when adapting either convolutional or attention layers. For convolutional adaptation, Tent collapses from 24.76 to 0.15, whereas ours improves from 36.61 to 39.26. For attention adaptation, Tent drops from 25.13 to 0.18, while ours increases from 42.15 to 45.94. These results suggest that MGP generalizes beyond normalization-only adaptation.
>
> > Q4: How reliable is the tracker under lower ER proportion, weaker source models, unfamiliar domains, or rapidly evolving directions?
>
> **A4.** Thank you for this valuable question. We added several stress tests to probe tracker reliability under these challenging conditions:
>
> - **Low ER proportion.** The blind-spot setting (Table 4, 0% initial accuracy) serves as an extreme low-reliable-signal case, yet our method remains stable with low tracking misalignment ($\delta \approx 0.25$), suggesting that the tracker remains reliable even when the ER signal is weak.
>
> - **Weaker source models.** As a proxy, we additionally tested a weaker backbone/source model (ResNet-18 instead of ResNet-50). MGP remains stable and improves over time on IN-C (34.86 $\to$ 35.25), while Tent collapses (31.13 $\to$ 1.37), indicating that MGP can still track a useful protected subspace even with a weaker source model. Detailed results are available **[here](https://anonymous.4open.science/r/tta-373C/fig_weak_source.png)**.
>
> - **Unfamiliar domains.** We also tested the two-stage abrupt-shift setting described above (**A2**): after 20 rounds on IN-C, we switch to a substantially different IN-K stream with an imbalanced label-distribution shift for another 20 rounds. Our method maintains gains within both stages (Stage 1, 53.25 $\to$ 55.69; Stage 2, 36.68 $\to$ 44.36), indicating robustness to unfamiliar domains and abrupt stream changes.
>
> - **Rapidly evolving directions / tracking lag.** We tested a much slower tracker refresh rate $K = 1000$, where MGP still attains R40 = 41.07 with $\delta \approx 0.28$. We also evaluated a fully shuffled-order stream on IN-C (ViT-B/16), where MGP still remains stable rather than collapsing (53.13 $\to$ 55.26), indicating that the tracker can still provide a useful protected subspace even under rapidly evolving directions and tracking lag. Detailed results are provided in Reviewer c9bv's response to Q3 and are also available **[here](https://anonymous.4open.science/r/tta-373C/fig_shuffled_streams.png)**.
>
> Overall, these results suggest that MGP degrades gracefully rather than catastrophically under severe low-signal, abrupt-shift, and tracking-lag stress conditions, which is consistent with our tracking-limited theory.
>
> *We thank you for appreciating our contributions. We sincerely hope the above clarifications have addressed your questions.*

---

> > ### Author Rebuttal · Reviewer_L1EY · 2026-04-01
> >
> > I appreciate the authors’ rebuttal effort and detailed explanations, and all of my concerns have been resolved. I am glad to see such solid work accepted by ICML.

---

> > > ### Author Response · Authors · 2026-04-02
> > >
> > > Dear Reviewer,
> > >
> > > We are glad to know that our response has addressed your concerns. Thank you again for your constructive feedback.
> > >
> > > Best,
> > >
> > > The Authors

---

### Official Review · Reviewer_DaRE · 2026-03-12

**Soundness:** 4
**Presentation:** 3
**Significance:** 3
**Originality:** 4
**Overall Recommendation:** 5
**Confidence:** 4

**Summary:**

This paper studies long-horizon test-time adaptation and attributes model collapse to spurious gradient leakage into a protected knowledge subspace. To address this issue, the authors propose Manifold-Aware Gradient Projection (MGP), which tracks the dominant gradient subspace online and projects adaptation gradients onto its orthogonal complement, aiming to prevent destructive updates while still allowing useful adaptation.

**Compliance With Llm Reviewing Policy:**

Affirmed.

**Final Justification:**

My concerns have been addressed. I keep my original score.

**Key Questions For Authors:**

For the parameter sensitivity, such as subspace rank and refresh frequency, could the authors provide more results on more datasets and models?

Could the proposed approach be combined with sample-selection, confidence-based, or consistency-based methods for further improvements?

**Limitations:**

yes

**Strengths And Weaknesses:**

++Pros

The paper introduces a novel geometric perspective on long-horizon TTA collapse, which is interesting and potentially insightful.

The proposed method is conceptually simple and straightforward.

Experiments on long-horizon revisit protocols and blind-spot settings demonstrate improved stability compared with several existing baselines.

The paper also provides additional analyses (e.g., spectral diagnostics and drift observations) to help support the proposed mechanism.


—Cons

The theoretical analysis relies on several strong assumptions and mainly serves as intuition rather than providing strict guarantees for deep models.

The experiments focus primarily on ImageNet-style classification benchmarks, which somewhat limits the demonstrated generality of the approach. Results on more datasets would be much better.

---

> ### Author Rebuttal · Authors · 2026-03-30
>
> Thank you for recognizing the novelty and contributions of our paper. We address your questions below.
>
> > Q1: For the parameter sensitivity, such as subspace rank and refresh frequency, could the authors provide more results on more datasets and models?
>
> **A1.** We additionally ran long-horizon sensitivity experiments on ImageNet-3DCC with ResNet-50 and ImageNet-R with ViT-B/16, covering subspace rank ($r\_{\max}$), refresh frequency ($K$), buffer size ($n$), and novelty threshold ($\tau\_{\mathrm{novel}}$). Across all settings on both dataset-model pairs, MGP consistently avoids collapse and remains stable over long horizons, demonstrating strong robustness to hyperparameter choices. Notably, even with very infrequent refreshes (e.g., $K=1000$) or relatively small buffers, MGP remains stable without collapse. Detailed results are available **[here](https://anonymous.4open.science/r/tta-373C/hyperparameters.png)**.
>
> > Q2: Could the proposed approach be combined with sample-selection, confidence-based, or consistency-based methods for further improvements?
>
> **A2.** Thank you for your valuable suggestion. We view these methods as complementary. If a filtering method provides a binary or soft weight $w_t$, the combined update becomes $\mathbf{P}_t^\perp (w_t g_t^{\mathrm{raw}})$. Filtering decides whether to update, while MGP decides where the admitted update can move.
>
> Below we report a preliminary experiment on ImageNet-C with ResNet-50, showing final accuracy at R40 using a representative entropy-gated variant following SAR [1]:
>
> | Method | Acc. |
> | --- | :---: |
> | Tent | 0.46 |
> | Tent + Filter ($\tau = 0.3$) | 0.10 |
> | Tent + Filter ($\tau = 1.0$) | 0.11 |
> | Ours | 42.96 |
> | Ours + Filter ($\tau = 0.3$) | 43.11 |
> | Ours + Filter ($\tau = 1.0$) | 43.01 |
>
> The main takeaway is compatibility: filtering can slightly improve which updates are admitted, while MGP addresses the orthogonal question of where those updates are allowed to move. Stricter gating may delay collapse, but by itself it does not prevent it.
>
> [1] Niu, S., et al. "Towards stable test-time adaptation in dynamic wild world." *ICLR*, 2023.
>
> *We appreciate your positive feedback on our contributions and hope these clarifications have addressed your questions.*

---

> > ### Author Rebuttal · Reviewer_DaRE · 2026-04-03
> >
> > Thanks to the authors for their response. The concerns I raised in the Questions section have been addressed. However, the concerns in my Weaknesses section were not addressed. While I do not consider these two points to be fundamental flaws, it is somewhat dismissive to directly ignore both of them, especially given that there appears to have been sufficient space to respond.  I keep my original score.

---

> > > ### Author Response · Authors · 2026-04-04
> > >
> > > Thank you for this helpful follow-up. We greatly appreciate your clarification, and we apologize that our previous response did not fully address the concerns you raised.
> > >
> > > On the theory side, our goal is to formalize a concrete mechanism that helps explain the long-horizon collapse studied in this work. Specifically, our analysis identifies persistent leakage into a protected low-rank subspace as one sufficient collapse channel, shows how this leakage can accumulate coherently over long horizons, and explains why MGP mitigates it. We therefore view this as a mechanism analysis of a concrete failure mode. In the revision, we will clarify this theoretical scope and positioning.
> > >
> > > To strengthen the empirical evidence beyond the ImageNet family, we additionally evaluate MGP on Camelyon17-v1 [1], a medical histopathology benchmark from WILDS [2] with cross-hospital distribution shift arising from differences in hospitals, patient populations, staining, and image acquisition. Under the same long-horizon protocol, Tent drops from 56.17 at R1 to 48.77 at R40 (-7.40), whereas MGP changes from 64.33 to 65.34 (+1.01). These results suggest that the long-horizon stabilization effect of MGP is not confined to ImageNet-style natural-image settings, but also extends to a representative medical benchmark under substantial distribution shift.
> > >
> > > Given the cost of long-horizon evaluation, it was not feasible to include broader validation beyond the ImageNet family within the rebuttal window. Nevertheless, we agree that broader validation beyond the ImageNet family is important for strengthening the empirical scope of the work. In the extended revision, we will include additional datasets to further broaden this empirical support.
> > >
> > > [1] Bandi et al. "From Detection of Individual Metastases to Classification of Lymph Node Status at the Patient Level: The CAMELYON17 Challenge." *IEEE Transactions on Medical Imaging*, 2019.
> > >
> > > [2] Koh et al. "WILDS: A Benchmark of in-the-Wild Distribution Shifts." *ICML*, 2021.

---

### Official Review · Reviewer_c9bv · 2026-03-13

**Soundness:** 3
**Presentation:** 4
**Significance:** 3
**Originality:** 3
**Overall Recommendation:** 5
**Confidence:** 3

**Summary:**

The paper addresses the challenge of instability and "model collapse" in long-horizon Test-Time Adaptation (TTA). While TTA allows pre-trained models to adapt online to distribution shifts, current methods often fail during prolonged deployments due to error accumulation. The authors identify a geometric failure mode called manifold erosion, where spurious gradients from confident mispredictions exhibit a persistent "leakage" into a low-rank subspace containing core model knowledge. To prevent this, they propose Manifold-Aware Gradient Projection (MGP), which tracks this dominant subspace online and projects updates onto its orthogonal complement, thereby blocking the leakage while preserving adaptation plasticity.

**Compliance With Llm Reviewing Policy:**

Affirmed.

**Final Justification:**

My concerns have been addressed, so I have decided to raise my score to 5.

**Key Questions For Authors:**

1. While your theoretical derivation for "manifold erosion" applies to general gradients, the empirical validation follows the TENT paradigm by only adapting normalization parameters. Since these layers have relatively low dimensionality, the spectral analysis is computationally efficient. How does MGP scale—both in terms of computational overhead and the stability of the subspace estimation—when applied to high-dimensional weights (e.g., convolutional or attention layers)?

2. MGP ensures stability by "locking" the core manifold and restricting all adaptation to the orthogonal residual space. In an open-ended, long-horizon deployment with a high diversity of domains, is there a risk of "adaptation saturation"? Essentially, if the model continues to encounter new shifts, could the residual space eventually lack the capacity to capture new features, leading to a performance plateau? Have you observed any signs of "gradient starvation" where the projected updates are no longer able to effectively minimize the test-time objective?

3. In the long-horizon cyclic experiments, MGP consistently achieves a positive $\Delta$, meaning accuracy actually improves over 40 rounds of revisits.  Does this suggest that the model is effectively "learning" a more robust representation of the common manifold across all corruptions? More importantly, if the stream were shuffled/randomized instead of cyclic, would MGP still show this cumulative improvement, or is the gain dependent on the repeated exposure to the same domain sequence?

**Limitations:**

yes

**Strengths And Weaknesses:**

Strengths:
- The paper moves beyond traditional "sample-based" filtering (which asks which samples to use) to a "parameter-based" geometric approach (which asks where the update should occur). This shift is highly significant for the TTA field, where heuristic filtering often fails.
- The authors provide a compelling analysis of why models collapse over long horizons. By demonstrating that spurious gradients exhibit "spectral asymmetry"—having high-rank noise but a biased projection onto the reliable low-rank manifold—they offer a rigorous explanation for why simple entropy or confidence filtering is insufficient.
- The authors provide a drift bound analysis showing that MGP keeps the parameter drift bounded, unlike unconstrained TTA which drifts linearly with time.
- Empirically, the proposed method consistently outperforms other baselines on diverse synthetic and natural shifts over long horizons, especially in a “blind-spot” regime, where the stream consists entirely of samples misclassified by the source model.

Weaknesses:
- While the spectral analysis is derived for general gradients, the empirical validation is largely restricted to normalization layers (following the TENT paradigm). It remains an open question whether the "manifold erosion" logic holds—or if the low-rank subspace estimation remains computationally and statistically feasible—when adapting high-dimensional weights in convolutional or attention layers.
- The method effectively "locks" the core manifold to prevent erosion. However, as adaptation progresses over increasingly diverse domains, the residual space (the only space where learning is allowed) may eventually become saturated or insufficient to capture new domain features, potentially leading to a performance plateau that the paper does not fully explore.

---

> ### Author Rebuttal · Authors · 2026-03-30
>
> Thank you for your valuable feedback and constructive comments on improving our paper. We address your questions below.
>
> > Q1: Why only normalization layers? How does MGP scale to higher-dimensional weights?
>
> **A1.** Our main experiments follow the standard Tent-family setup for a fair comparison, and long-horizon collapse is already clear in that setting. However, MGP is not restricted to normalization layers: the per-step projection cost is $\mathcal{O}(d_a r_t)$ and the refresh cost is $\mathcal{O}(n d_a r_t)$, so the overhead scales with the active parameter subset size $d_a$, not the total model size.
>
> We also ran broader-subset experiments on ImageNet-C (IN-C):
>
> |Backbone|Adapted params|Method|R1|R40|$\Delta$|Peak GPU Memory (GB)|Time / 1k imgs|
> |---|---|:---:|:---:|:---:|:---:|:---:|:---:|
> |ResNet-50|Default (norm)|Tent|37.51|0.46|-37.05|8.21|1.30|
> |ResNet-50|Default (norm)|Ours|41.90|42.96|+1.06|13.36|4.28|
> |ResNet-50|Conv|Tent|24.76|0.15|-24.61|8.37|3.34|
> |ResNet-50|Conv|Ours|36.61|39.26|+2.65|13.99|5.52|
> |ViT-B/16|Attention|Tent|25.13|0.18|-24.95|11.14|4.44|
> |ViT-B/16|Attention|Ours|42.15|45.94|+3.79|18.02|7.89|
>
> The stability advantage therefore persists beyond normalization-only adaptation, while the overhead grows naturally with the size of the active subset.
>
> > Q2: Could the residual space eventually saturate? Any sign of gradient starvation?
>
> **A2.** Thank you for your valuable question. Within the horizons we study, we do not observe residual-space saturation or gradient starvation. On IN-C, the projected gradient energy ratio $\rho_t=\|\mathbf{P}_t^\perp g_t^{\mathrm{raw}}\|_2 / \|g_t^{\mathrm{raw}}\|_2$ remains stable over time (R1 / R40: 0.56 / 0.53), and the effective residual rank stays around 25.
>
> To probe this more directly, we ran a two-stage stress test on ViT-B/16: Stage 1 adapts for 20 rounds over 15 IN-C domains; Stage 2 then switches to a substantially different ImageNet-K (IN-K) stream with an imbalanced label distribution shift simulated by a Dirichlet distribution ($\beta = 0.1$) for another 20 rounds:
>
> |Method|Stage 1 (IN-C, R1)|Stage 1 (IN-C, R20)|Stage 2 (IN-K, R1)|Stage 2 (IN-K, R20)|
> |---|:---:|:---:|:---:|:---:|
> |Tent|36.89|0.10|0.24|0.21|
> |ETA|49.61|0.11|0.44|0.42|s
> |CoTTA|40.34|41.31|30.40|30.48|
> |DeYO|14.29|0.16|0.11|0.11|
> |PTTA|24.67|0.12|0.05|0.04|
> |SPA|49.56|0.09|0.13|0.13|
> |Ours|**53.25**|**55.69**|**36.68**|**44.36**|
>
> Most baselines collapse during Stage 1 and remain near zero after the abrupt Stage-2 shift. In contrast, our method remains robust across both stages, improving in Stage 1 (53.25 $\to$ 55.69) and continuing to improve in Stage 2 (36.68 $\to$ 44.36), suggesting that the available update space is not exhausted within the horizons we study.
>
> We also ran an extreme single-domain saturation test (20 rounds on one domain, then evaluation on the remaining 14 domains):
>
> |Metric|Tent|ETA|CoTTA|RoTTA|SAR|DeYO|TRIBE|COME|PTTA|SPA|Ours|
> |---|:---:|:---:|:---:|:---:|:---:|:---:|:---:|:---:|:---:|:---:|:---:|
> |Single-domain acc. (R20)|16.27|37.93|10.27|11.16|27.25|27.71|0.44|26.04|33.74|15.65|**42.51**|
> |Remaining-domain avg. acc.|27.53|33.87|33.17|1.38|33.64|27.79|0.18|31.53|33.40|12.48|**37.15**|
>
> The same conclusion holds: MGP shows no sign of residual-space exhaustion under prolonged adaptation. Detailed per-domain results: **[link](https://anonymous.4open.science/r/tta-373C/fig_saturation.png)**.
>
> > Q3: MGP consistently achieves a positive $\Delta$ in cyclic experiments. Does this suggest the model is effectively "learning" a more robust representation of the common manifold? Would this remain true under shuffled/randomized streams?
>
> **A3.** Thank you for your valuable question. The positive $\Delta$ in the cyclic experiments suggests that MGP can accumulate beneficial updates across recurring corruption patterns without destabilizing the model. To test whether this depends on a fixed cyclic order, we additionally evaluated a shuffled-order stream, where domains are randomly permuted at every round (IN-C, ViT-B/16):
>
> |Metric|Tent|ETA|CoTTA|SAR|DeYO|COME|SPA|PTTA|Ours|
> |---|:---:|:---:|:---:|:---:|:---:|:---:|:---:|:---:|:---:|
> |Acc. (R1)|35.80|49.62|40.42|48.33|10.29|45.06|50.06|2.92|**53.13**|
> |Acc. (R40)|0.10|0.11|40.37|47.29|0.16|0.12|0.14|0.13|**55.26**|
> |$\Delta$|-35.70|-49.51|-0.05|-1.04|-10.13|-44.94|-49.92|-2.79|**+2.13**|
>
> As shown above, our method remains stable and continues to improve (53.13 $\to$ 55.26), indicating that the positive gain is not tied to a fixed domain order. More broadly, the result suggests that MGP can accumulate useful updates over time without destabilizing drift.
>
> *We sincerely hope our clarifications above have addressed your concerns and can improve your opinion of our work.*

---

> > ### Author Rebuttal · Reviewer_c9bv · 2026-04-02
> >
> > I appreciate the detailed explanations and additional empirical results. My concerns have been addressed, so I have decided to raise my score to 5.

---

> > > ### Author Response · Authors · 2026-04-02
> > >
> > > Dear Reviewer,
> > >
> > > Thank you for taking the time to review our rebuttal. We sincerely thank you again for your constructive feedback and for raising your score.
> > >
> > > Best,
> > >
> > > The Authors

---

### Official Review · Reviewer_2iuV · 2026-03-16

**Soundness:** 2
**Presentation:** 3
**Significance:** 3
**Originality:** 2
**Overall Recommendation:** 5
**Confidence:** 3

**Summary:**

The authors propose a novel method for test-time adaptation. It is based on certain geometric observations of how modern classifiers handle test-time distribution shift: e.g. that _collapse is associated with a persistent projection of spurious gradients into a protected, low-rank knowledge subspace, whose coherent accumulation erodes core representations over time._ The authors argue that by projecting into this subspace, the degree to which spurious updates harm test-time adaptation can be reduced. They validate their model on a variety of datasets and compare to a large number of competitive models.

**Compliance With Llm Reviewing Policy:**

Affirmed.

**Final Justification:**

Authors have addressed my technical concerns.

**Key Questions For Authors:**

Please see weaknesses.

**Limitations:**

yes

**Strengths And Weaknesses:**

**Strengths**
Overall, the paper is strong with the ideas and experimental validation.
- The paper is clear and technically comprehensive. The experimental validation is thorough.
- The idea is well-motivated.
- The code is released, which is greatly appreciated.

**Weaknesses**
Currently, I am split on this paper since the experimental results are so good. But I list the paper as a reject / weak reject due to issues with clarity and a few potential problems with the proofs. However, I would be happy to discuss with the authors.
- There are some issues with the clarity of certain technical statements. I list a few below. Could the authors please clarify them?
- The paper seems largely edited by LLM tools. This is not necessarily a bad thing, but in this case it is quite jarring at least for me to read since the editing seems extreme. If I am being honest, I don't like it used to this degree, since it makes me (and potentially other readers) distrustful of some material. For example, in Definition 3.1, the condition seems like a correctness condition, but the authors characterize the confidence of the model just using the sign of the margin which seems strange to me. (my guess is that the authors mean that the confidence should be related to the magnitude of the margin, but the LLM added the parenthetical?). Additionally, there are strange notational artifacts like "bulk-edge" separation (btw, I have never seen this term used) used only a few times that seem to just add additional notations that are not needed.
- There are some missing citations - e.g. the ideas from multitask learning seem relevant. For example, I googled 1 paper on gradient surgery for TTA, which seems very relevant.

**Technical statements confusion**
There are possibly others, but here are some potential mistakes I found.
- Notations are used liberally without proper introduction, making it very hard to read this paper as a human being. E.g. g_t^{ES}, g_t^{ER}, etc. The meaning of these notations can be inferred, but it is not ideal as a reader.
- I am confused by the proposition 3.3. (2) in the main text identifies g_t as g_t = g_t^{ES} + g_t^{ER}, but it seems the ER term is dropped in the proof in the appendix? This is not clear to me - I am not sure if it is a problem with my own understanding or the proof. I expect it is a  mistake.

---

> ### Author Rebuttal · Authors · 2026-03-30
>
> Thank you for taking the time to review our paper and for your detailed comments. We address your questions below.
>
> > Q1. The notations (e.g., "bulk-edge" separation, $g\_t^{\mathrm{ES}}$, $g\_t^{\mathrm{ER}}$) are used liberally without proper introduction.
>
> **A1.** Thank you for your question. By "bulk-edge" separation, we mean the standard spiked-covariance / Marchenko-Pastur intuition of separating an approximately isotropic bulk from low-rank spike directions: Eq. (5)–(6) estimate the MP bulk edge and retain only directions above it, with supporting details in Appendix D.8 and Appendix E. We will add the Marchenko-Pastur citation [1] at first use and introduce $g_t^{\mathrm{ER}}$ and $g_t^{\mathrm{ES}}$ explicitly before they appear.
>
> > Q2. In Definition 3.1, the condition seems like a correctness condition, but the confidence of the model is characterized just using the sign of the margin, which seems strange.
>
> **A2.** We agree that the current wording can be misleading. Definition 3.1 is not intended to define model confidence from the sign of PAM. In our setup, confidence for filtering-based TTA enters through the admission gate $\mathbb{I}_t \in \{0,1\}$ in Eq. (1). PAM is introduced only for labeled analysis: among admitted samples ($\mathbb{I}_t=1$), $\mathrm{PAM}>0$ corresponds to the correctness event and $\mathrm{PAM}\le 0$ to the incorrectness event. We use only this sign-based oracle partition to define ER and ES for analysis; the magnitude of PAM is not used as a confidence score. We will revise the wording to make this explicit.
>
> > Q3. I am confused by Proposition 3.3. The main text identifies $g\_t = g\_t^{\mathrm{ES}} + g\_t^{\mathrm{ER}}$, but it seems the ER term is dropped in the proof in the appendix?
>
> **A3.** The statement used in Proposition 3.3 is narrower than this interpretation suggests. The equation $g_t = g_t^{\mathrm{ES}} + g_t^{\mathrm{ER}}$ is an oracle bookkeeping split, rather than the projector decomposition in Eq. (2). The main-text decomposition is $g\_t \= \mathbf{P}\_{\mathcal{S}\_{\mathrm{know}}} g\_t + \mathbf{P}^{\perp}\_{\mathcal{S}\_{\mathrm{know}}} g\_t,$ which is a projector decomposition of the full update.
>
> The ER/ES split is used only in Sec. 3 as an oracle diagnostic to isolate one sufficient collapse channel. Concretely, the intended sufficient-mechanism statement is to isolate the ES-induced protected component $\Delta\_T^{\mathrm{ES}} := -\eta \sum\_{t<T} \mathbb{I}\_t \mathbf{P}\_{\mathcal{S}\_{\mathrm{know}}} g\_t^{\mathrm{ES}}$, for which $\mathbb{E}[\Delta\_T^{\mathrm{ES}}] = -\eta \sum\_{t<T}\mu\_t$, where $\mu\_t := \mathbb{E}\left[\mathbb{I}\_t \mathbf{P}\_{\mathcal{S}\_{\mathrm{know}}} g\_t^{\mathrm{ES}}\right]$.
>
> Hence, if $\mu_t \equiv \mu \neq 0$, the ES-induced protected drift grows linearly with $T$. Appendix D.6 then makes the asymptotic point explicit: a persistent bias in protected directions yields $\Theta(T)$ mean drift, whereas stochastic deviation scales only as $O(\sqrt{T})$. Therefore, even a small coherent ES leakage can dominate over long horizons.
>
> So the intended claim is **not** that all protected drift comes only from ES. Rather, the ES component alone already provides a sufficient collapse mechanism. We will revise the wording to clarify this point.
>
> Importantly, the actual method and the main guarantee in Sec. 4 are defined directly on the full unlabeled raw gradient ($g_t^{\mathrm{raw}}$) through the projector update $g_t^{\mathrm{safe}}=\mathbf{P}_t^\perp g_t^{\mathrm{raw}}$, and do not depend on observing the oracle ER/ES split at test time.
>
> > Q4. There are some missing citations - e.g. the ideas from multitask learning seem relevant.
>
> **A4.** Thank you for your suggestion. We will add representative references on gradient surgery, including PCGrad [2] and a recent TTA application [3]. That said, the setting differs from ours: such methods resolve conflicts among multiple known gradients (e.g., from different tasks, prototypes, or auxiliary objectives) by removing or projecting away conflicting components. MGP instead adapts a single model to a continuous unlabeled test stream. Moreover, gradient surgery mainly addresses pairwise conflicts, while MGP tracks a low-rank protected subspace from the stream and projects each update onto its orthogonal complement to prevent manifold erosion.
>
> [1] Marchenko, V. A., & Pastur, L. A. "Distribution of eigenvalues for some sets of random matrices." *Mathematics of the USSR-Sbornik*, 1967.
>
> [2] Yu, T., et al. "Gradient Surgery for Multi-Task Learning." *NeurIPS*, 2020.
>
> [3] Shamsi, A., et al. "Gradient Surgery: A Necessity for Robust Test-Time Adaptation for Detecting Casting Defects." *Engineering Applications of Artificial Intelligence*, 2025.
>
> *We sincerely hope the above clarifications have addressed your concerns. Kindly let us know if you have any further comments, as we would be happy to continue the discussion.*

---

> > ### Author Rebuttal · Reviewer_2iuV · 2026-04-02
> >
> > Thanks for the detailed response.. particularly the clarifications around the bulk-edge notation, Definition 3.1 and the additional references . Most of my concerns have been addressed, but I would like to resolve some lingering confusion about Proposition 3.3 and it's proof in the appendix.
> >
> > This collapse / domination explanation in the rebuttal is a little hard for me to grasp. In particular, you mention that there is some additional material in appendix D.6 that I am missing, but I don't see an appendix D.6 in the paper? Do you mean C.6?
> >
> > So if I understand correctly, there are two decompositions: the projector decomposition in (2) and this bookkeeping split, where gradients are classified by whether the sample is ER or ES. In prop 3.3 in the main text, expectation of \Delta_t is defined as some sum over the projected ES gradients. But you say \Delta_t in the prop 3.3 proof first line in the appendix is defined by that telescoped term. Using (2), the split expands into a term involving ER and ES. Where does the ER gradient term disappear to?
> >
> > I can maybe see from C.6 how the result that ES component can dominate a stochastic term is motivational, but I don't see either justifications: that the ER gradients are dominated or in expectation they are negligible. In fact, the authors define the subspace S_{know} in terms of the covariance of the ER gradients which seems contradictory. Am I missing something critical?

---

> > > ### Author Response · Authors · 2026-04-03
> > >
> > > Thank you for the careful follow-up. We agree that the current presentation can be read as conflating two different quantities: the total protected drift and the ES-induced protected component. Our intent in Proposition 3.3 was to isolate the ES-induced protected component as a sufficient collapse channel arising from persistent ES leakage, rather than to claim that the ER contribution disappears from the total protected drift.
> > >
> > > > Q1. Where does the ER gradient term disappear to?
> > >
> > > **A1.** It is still part of the total protected drift. Let $\Delta\_T := \mathbf P\_{\mathcal S\_{\mathrm{know}}}(\theta\_T-\theta\_0)$ denote the total protected drift.
> > > Using the oracle split on admitted updates, we define $g\_t^{\mathrm{ES}} := \mathbb{J}\_t g\_t,$ and $g\_t^{\mathrm{ER}} := (1-\mathbb{J}\_t) g\_t,$ where $\mathbb{J}_t=1$ if the admitted sample at step $t$ is ES, and $\mathbb{J}_t=0$ otherwise. This yields $g_t = g_t^{\mathrm{ER}} + g_t^{\mathrm{ES}}$. Therefore,
> > >
> > > $$
> > > \Delta\_T
> > > \=
> > > -\eta\sum\_{t<T}\mathbf P\_{\mathcal S\_{\mathrm{know}}} g\_t^{\mathrm{ER}}
> > > +
> > > -\eta\sum\_{t<T}\mathbf P\_{\mathcal S\_{\mathrm{know}}} g\_t^{\mathrm{ES}}.
> > > $$
> > >
> > > So the ER term remains part of the total drift.
> > >
> > > The intended statement of Proposition 3.3 is narrower: to isolate the ES-induced protected component $\Delta_T^{\mathrm{ES}} := -\eta\sum_{t<T}\mathbf P_{\mathcal S_{\mathrm{know}}} g_t^{\mathrm{ES}},$ and show that $\mathbb E[\Delta_T^{\mathrm{ES}}] \= -\eta\sum_{t<T}\mathbb E[\mathbf P_{\mathcal S_{\mathrm{know}}} g_t^{\mathrm{ES}}].$ In particular, if this protected-direction ES bias is persistent and nonzero, then the expected ES-induced protected drift grows linearly with $T$.
> > >
> > > Thus, the intended claim is a sufficiency statement: even without assuming the ER term is negligible, the ES component alone already provides a linear-drift collapse channel. We agree that the current wording should make this distinction explicit.
> > >
> > > > Q2. Why is this not contradictory to defining $\mathcal S_{\mathrm{know}}$ from ER gradients?
> > >
> > > **A2.** There is no contradiction because these concern different moments of the gradient distribution.
> > >
> > > - $\mathcal S_{\mathrm{know}}$ is defined from the dominant eigenspace of the second moment of ER gradients,
> > >   $\mathbb E[g^{\mathrm{ER}}(g^{\mathrm{ER}})^\top],$ which identifies directions where reliable gradients have concentrated energy.
> > >
> > > - The collapse channel concerns the first-moment bias of ES projections into that same subspace, $\mu^{\mathrm{ES}} := \mathbb E[\mathbf P_{\mathcal S_{\mathrm{know}}} g^{\mathrm{ES}}].$
> > >
> > > So ER gradients can have large energy in $\mathcal S_{\mathrm{know}}$ while contributing zero or benign mean drift, whereas ES gradients can have a smaller but persistently biased projection into $\mathcal S_{\mathrm{know}}$. Appendix C.6 (previously mis-cited as D.6) is intended only to support this bias-vs-fluctuation point: once such a protected-direction bias exists, its mean accumulation is linear in $T$, while stochastic fluctuation is $O(\sqrt{T})$.
> > >
> > > > Q3. How will this be clarified in the revision?
> > >
> > > **A3.** We will revise the statement around Proposition 3.3 to explicitly distinguish the total protected drift from the ES-induced protected component, and to make clear that the argument in Sec. 3 isolates the latter as a sufficient mechanism rather than asserting that the ER contribution vanishes.
> > >
> > > This interpretation is also consistent with our oracle diagnostics: in Fig. 3(a), removing protected-subspace ER projections has little effect on adaptation, whereas in Fig. 3(c), ES leakage closely tracks degradation. We view this as empirical support for the sufficient-mechanism statement above, rather than as an assumption used in Proposition 3.3.
> > >
> > > Importantly, this clarification only affects the oracle mechanism discussion in Sec. 3. The actual MGP update and the Sec. 4 drift guarantees are defined directly on the full raw gradient $g\_t^{\mathrm{safe}}=\mathbf P\_t^\perp g\_t^{\mathrm{raw}},$ and do not depend on observing the ER/ES split at test time.
> > >
> > > *We hope these revisions and clarifications have alleviated your remaining concerns and will lead to a more positive assessment of our work.*
> > >
> > > ---
> > >
> > > *Additional note: Thank you for your continued engagement and constructive feedback. We truly appreciate your time and your decision to raise the score.*

---

### Decision · Program_Chairs · 2026-04-30

**Decision:**

Accept (regular)

**Comment:**

Test-time adaptation updates during testing to reduce generalization on shifted data. A particularly difficult setting is that of long-horizon adaptation where long means a larger amount of data and therefore more updates. The proposed manifold-aware gradient projection (MGP) method is a subspace optimization method that improves the stability of adaptation in this setting and prevents the "collapse" of predictions such that error increases rather than reduces.

All reviewers agree on acceptance given the strong empirical results. While there were initial concerns about technical correctness, these were solved in the rebuttal, and acknowledged by the reviewers. 2iuV confirms the clarifications about the derivations and raises the score. c9bv confirms the relevance and quality of the additional empirical results on parameterizations, the robustness of optimization over sufficient time and different data as well as recurring passes over the same data, and the improvement of MGP even for random domain orders. DaRE first confirms the answers to their questions but notes the lack of response to the identified weaknesses, but these are then partially resolved by a follow-up with clarification of theory and more results on another dataset and promise of more datasets in the revision. L1EY confirms their satisfaction with the convincing experiments as augmented by the rebuttal and its coverage of memory usage, non-stationary shift, and more adaptation conditions. Given the positive consensus, and the absence of remaining obstacles, the area chair sides with acceptance.

Note: The area chair recognizes the confidential comments by the authors. The discussion between reviewers and authors is incorporated into the decision in its full detail.